# Towards Parameter-Free Temporal Difference Learning

**Yunxiang Li** [1] **Mark Schmidt** [1] **Reza Babanezhad** [2] **Sharan Vaswani** [3]

## Abstract

Temporal difference (TD) learning is a fundamental algorithm for estimating value functions in reinforcement learning. Recent finite-time analyses of TD with linear function approximation quantify its theoretical convergence rate. However, they often require setting the algorithm parameters using problem-dependent quantities that are difficult to estimate in practice — such as the minimum eigenvalue of the feature covariance ($\omega$) or the mixing time of the underlying Markov chain ($\tau_{\mathrm{mix}}$). In addition, some analyses rely on nonstandard and impractical modifications, exacerbating the gap between theory and practice. To address these limitations, we use an exponential step-size schedule with the standard TD(0) algorithm. We analyze the resulting method under two sampling regimes: independent and identically distributed (i.i.d.) sampling from the stationary distribution, and the more practical Markovian sampling along a single trajectory. In the i.i.d. setting, the proposed algorithm does not require knowledge of problem-dependent quantities such as $\omega$, and attains the optimal bias-variance trade-off for the last iterate. In the Markovian setting, we propose a regularized TD(0) algorithm with an exponential step-size schedule. The resulting algorithm achieves a comparable convergence rate to prior works, without requiring projections, iterate averaging, or knowledge of $\tau_{\mathrm{mix}}$ or $\omega$.

## 1. Introduction

Reinforcement learning (RL) is a general framework for sequential decision making under uncertainty, with successes in robotics (Kober et al., 2013) and in aligning language models (Uc-Cetina et al., 2023). Value functions underpin value-based algorithms (Sutton & Barto, 2018) and are

central to actor-critic methods (Konda & Tsitsiklis, 1999), making efficient policy evaluation a core RL task.

Temporal-difference (TD) learning (Sutton, 1988) is an incremental policy-evaluation method that bootstraps value estimates and scales with linear function approximation. While convergence of TD with linear function approximation (Tsitsiklis & Roy, 1997; Dalal et al., 2018; Lakshminarayanan & Szepesvári, 2018; Mou et al., 2020; Bhandari et al., 2018; Patil et al., 2023; Samsonov et al., 2024) and its variants (Liu & Olshevsky, 2021; Mustafin et al., 2024) have been analyzed, many analyses require hard-to-estimate problem-dependent parameters or nonstandard modifications (e.g., projections or iterate averaging). We therefore seek to *design a theoretically principled TD algorithm that requires only minimal modifications and does not rely on knowledge of problem-dependent constants.*

To that end, we first consider the independent and identically distributed (i.i.d.) sampling regime, where states are sampled from the stationary distribution of the underlying Markov chain for the evaluated policy. The i.i.d. sampling regime is often used as a testbed for designing and analyzing policy-evaluation algorithms (Lakshminarayanan & Szepesvári, 2018; Dalal et al., 2018; Bhandari et al., 2018; Patil et al., 2023; Samsonov et al., 2024).

Dalal et al. (2018) analyze the TD(0) algorithm using tools from stochastic approximation. Their choice of step-size trades off bias (the rate at which the initialization is forgotten) and variance (from i.i.d. sampling) for the last iterate. Follow-up work by Bhandari et al. (2018) relates TD to stochastic gradient descent (SGD) and uses optimization tools to analyze TD. They study three step-size schedules (see Table 1) under i.i.d. sampling. Some require knowledge of the smallest eigenvalue of the state-weighted feature covariance $\omega$, while others yield slower rates. While these rates hold for the last iterate, they do not achieve the optimal bias–variance trade-off. More recent work (Patil et al., 2023; Samsonov et al., 2024) adopts stochastic approximation, a different technique from ours, and uses tail averaging to achieve the optimal bias–variance trade-off without problem-dependent constants. However, we note that typical practical implementations of TD do not use iterate averaging.

[1] University of British Columbia [2] Samsung AI - Montreal [3] Simon Fraser University. Correspondence to: Yunxiang Li <liyx12@student.ubc.ca>.

*Proceedings of the 43$^{rd}$ International Conference on Machine Learning*, Seoul, South Korea. PMLR 306, 2026. Copyright 2026 by the author(s).

| Sampling | Step-size | Convergence rate | Parameters needed | Projection | Last or Average iterate convergence |
|---|---|---|---|---|---|
| i.i.d. | $O(t+1)^{-z}, z \in (0,1)$ (Dalal et al., 2018) | $O\big(\exp(-\omega T^{1-z}) + 1/T^z\big)$ | None | No | Last |
| | $1/\sqrt{T}$ (Bhandari et al., 2018) | $O\big(\frac{\sigma^2}{\sqrt{T}}\big)$ | None | No | Average |
| | $O(\omega)$ (Bhandari et al., 2018) | $O\big(\exp(-\omega^2 T) + \sigma^2\big)$ | $\omega$ | No | Last |
| | $O\big(\frac{1}{1+t\,\omega}\big)$ (Bhandari et al., 2018) | $O\big(\frac{\sigma^2}{T\omega}\big)$ | $\omega$ | No | Last |
| | $O(1)$ (Samsonov et al., 2024) | $\tilde{O}\big(\exp(-\omega T) + \frac{\sigma^2}{\omega^2 T}\big)$ | None | No | Average |
| | $O\big(\big(\frac{1}{T}\big)^{t/T}\big)$ (**Ours**) | $\tilde{O}\big(\exp(-\omega T) + \frac{\sigma^2}{\omega^2 T}\big)$ | None | No | Last |
| Markovian samples | $1/\sqrt{T}$ (Bhandari et al., 2018) | $O\Big(\frac{\big(1+\tau_{\mathrm{mix}}(1/\sqrt{T})\big)}{\omega^2\sqrt{T}}\Big)$ | No | Yes | Average |
| | $O(1/\omega)$ (Bhandari et al., 2018) | $O(\exp(-2\eta\omega\,T)) + O\Big(\frac{\eta\big(1+\tau_{\mathrm{mix}}(\eta)\big)}{\omega^3}\Big)$ | $\omega$ | Yes | Last |
| | $O(1/(\omega\,(t+1)))$ (Bhandari et al., 2018) | $O\Big(\frac{\big(1+\tau_{\mathrm{mix}}(\alpha_T)\big)}{\omega^3} \cdot \frac{1+\log T}{T}\Big)$ | $\omega$ | Yes | Average |
| | $O(1)$ (Srikant & Ying, 2019) | $\tilde{O}(\exp(-\eta\omega T) + \eta\tau_{\mathrm{mix}})$ | $\tau_{\mathrm{mix}}$ | No | Last |
| | $O(1)$ for TD with data drop (Samsonov et al., 2024) | $\tilde{O}\big(\exp(-\omega T) + \frac{\tau_{\mathrm{mix}}}{\omega^2 T}\big)$ | $\tau_{\mathrm{mix}}$ | No | Average |
| | $O\big(\frac{\omega}{\tau_{\mathrm{mix}}}\big)$ (Mitra, 2025) | $O\big(\exp\big(-\frac{\omega^2(T+1)}{\tau_{\mathrm{mix}}}\big)\big) + \tilde{O}\big(\frac{\tau_{\mathrm{mix}}}{\omega^2(T+1)}\big)$ | $\tau_{\mathrm{mix}}, \omega$ | No | Average |
| | $\alpha/k^\xi, \xi \in (0.5,1)$ (Haque et al., 2024) | $\tilde{O}\big(\frac{\sigma^2}{T}\big)$ | None | No | Average |
| | $O\big(\frac{\omega}{\ln^2 T}\big(\frac{1}{T}\big)^{t/T}\big)$ (**Ours**) | $O\big(\exp\big(-\frac{\omega T}{\ln^3(T)}\big) + \frac{\ln^4(T)}{\omega^2 T}\exp\big(\frac{m}{\ln(1/\rho)}\big)\big)$ | $\omega$ | No | Last |
| | $O\big(\frac{1}{\sqrt{T}\,\ln^2 T}\big(\frac{1}{T}\big)^{t/T}\big)$ for regularized TD (**Ours**) | $O\big(\exp\big(-\frac{\omega\sqrt{T}}{\ln^3(T)}\big) + \frac{\ln^3(T)}{\omega^2 T}\exp\big(\frac{m}{\ln(1/\rho)}\big)\big)$ | None | No | Last |

*Table 1.* Comparison of our method and other methods. $\omega$ is the smallest eigenvalue of the state feature covariance matrix, $T$ is the number of updates, $\tau_{\mathrm{mix}}$ is the mixing time defined in Equation (3), and $m, \rho$ are constants in Definition 4.1, Our i.i.d. result and regularized TD(0) under Markovian sampling require no projections, no prior knowledge of $\tau_{\mathrm{mix}}$ or $\omega$, and no iterate averaging.

**Contribution 1**. For TD(0) with linear function approximation under i.i.d. sampling, we take an optimization lens similar to Bhandari et al. (2018) and develop a TD algorithm that uses exponentially decaying step-sizes (Li et al., 2021). This schedule was introduced by Li et al. (2021) for SGD on smooth, strongly convex objectives under i.i.d. noise, where standard strong convexity drives the contraction (Vaswani et al., 2022). However, the TD(0) update does not fit this framework: it is a semi-gradient, not the gradient of any fixed objective, and the key contraction operates in the value-space $\|\cdot\|_D$-norm via a one-point strong-monotonicity property (Lemma 3.1) rather than standard strong convexity.

We therefore develop a new analysis that uses this monotonicity in place of strong convexity, we are the first to prove that TD(0) with exponentially decaying step-sizes achieves the optimal bias-variance trade-off for the last iter-

ate, and does not require knowledge of problem-dependent constants such as $\omega$ (Section 3). We note that establishing last-iterate convergence is important because (a) it is used by typical practical implementations of TD and (b) maintaining a running average doubles the value-model's memory footprint which is non-trivial in large-scale RL. Furthermore, compared to average iterate convergence, it is deriving last-iterate guarantees is more difficult and requires different technical tools.

Since obtaining direct access to the stationary distribution is unrealistic, the i.i.d. regime is impractical. Consequently, many theoretical works analyze TD(0) with Markovian sampling (Bhandari et al., 2018; Samsonov et al., 2024; Patil et al., 2023; Mou et al., 2020; Chandak & Borkar, 2025). In this setting, data are collected along a single Markovian trajectory, introducing temporal dependence that complicates analysis. To enable analysis, prior work often assumes fast

mixing so the state distribution approaches stationarity exponentially quickly. Under this assumption, Bhandari et al. (2018) analyze a projected variant of TD(0) under three step-size schedules. Similar to the i.i.d. case, the algorithm does not achieve the optimal trade-off between bias and dependence on the mixing time. Moreover, the projection step is nonstandard in practice and requires knowledge of $\omega$. Chandak & Borkar (2025) treat TD(0) as a contractive stochastic approximation algorithm. However, they require $\omega$ to set the initial step-size and guarantee convergence only to a neighborhood of the solution.

Other recent Markovian analyses fall into two categories: (i) Samsonov et al. (2024); Patil et al. (2023) study TD with data drop, a nonstandard variant that does not explicitly analyze consecutive-sample correlations. Moreover, these methods discards samples, making them sample-inefficient and unlikely to be used in practice. (ii) Srikant & Ying (2019); Mitra (2025); Haque et al. (2024) control correlations between consecutive samples and prove convergence without projection, achieving the optimal bias–mixing time trade-off. Both Srikant & Ying (2019); Mitra (2025) require the knowledge of $\tau_{\mathrm{mix}}$. On the other hand, Haque et al. (2024) does not require the knowledge of $\tau_{\mathrm{mix}}$ but their convergence guarantees only hold for the average-iterate.

**Contribution 2**. In the Markovian sampling regime, we show that standard TD(0) with linear function approximation and exponentially decaying step-sizes achieves the optimal bias–mixing time trade-off. The algorithm requires neither projections, iterate averaging, nor data drop, and it does not require the mixing time (Section 4.1); however, it still depends on $\omega$. We remove this dependence by analyzing a regularized TD(0) variant (Patil et al., 2023) with exponentially decaying step-sizes. Unlike Patil et al. (2023), who use regularization to improve constants, we use it to make the algorithm parameter-free from problem-dependent constants. Our result remains parameter-free while retaining the benefits of standard TD(0) (Section 4.2).

Table 1 compares our results with prior work by convergence rate, required parameters, projection, and whether average- or last-iterate convergence is guaranteed. The rest of the paper is organized as follows: Section 2 formalizes the problem and notation and introduces TD(0) with exponentially decaying step-sizes. Section 3 presents the i.i.d. analysis. Section 4 extends the analysis to Markovian sampling. Section 4.2 establishes our parameter-free regularized TD(0) guarantees. In Section 5, we report experiments on a controlled synthetic environment (full setup in Appendix H).

## 2. Problem Formulation

In this section, we formalize the setting and notation, including the Markov decision process (MDP) and TD(0),

linear value-function approximation and assumptions, and an exponential step-size schedule.

**Markov decision process.** We consider a discounted MDP $M = (\mathcal{S}, \mathcal{A}, \pi, P_\pi, \mu_0, r, \gamma)$, where $\mathcal{S}$ is the state space, $\mathcal{A}$ is the action space, $\pi$ is a fixed policy mapping each state $s \in \mathcal{S}$ to a distribution over actions in $\mathcal{A}$, $P_\pi \in \mathbb{R}^{|\mathcal{S}| \times |\mathcal{S}|}$ is the transition matrix induced by $\pi$ with entries $(P_\pi)_{ij} \triangleq \mathbb{P}(s_{t+1} = s_j \mid s_t = s_i)$, $\mu_0$ is the initial state distribution, $r : \mathcal{S} \times \mathcal{A} \to \mathbb{R}$ is the reward function, and $\gamma \in (0, 1]$ is the discount factor. At time $t$, an action $a_t \sim \pi(\cdot \mid s_t)$ is selected, the next state $s_{t+1} \sim P_\pi(\cdot \mid s_t)$ is sampled, and a reward $r(s_t, a_t)$ is received. Because the policy $\pi$ is fixed, we define the expected immediate reward as $r(s) \triangleq \mathbb{E}_{a \sim \pi(\cdot \mid s)}[r(s, a)]$. For simplicity, we assume $r(s) \in [0, 1]$. Iterating this interaction produces a trajectory $\tau = (s_0, a_0, s_1, \dots)$ with distribution $p_\pi(\tau)$ under policy $\pi$. Let $\mu_\pi$ denote the stationary state distribution induced by $\pi$. The initial state distribution $\mu_0$ may differ from $\mu_\pi$. The state distribution at time $t$ is $P_\pi^t \mu_0$. The value function $V^\pi$ gives the expected cumulative discounted reward starting from $\mu_0$ and following policy $\pi$, i.e., $V^\pi(s) = \mathbb{E}_{\tau \sim p_\pi(\cdot \mid \mu_0)} \left[ \sum_{t=0}^\infty \gamma^t r(s_t) \right]$.

We consider three regimes for sampling: mean-path, i.i.d., and Markovian. In the mean-path setting, expectations are evaluated exactly under $\mu_\pi$. In the i.i.d. setting, samples are independent draws from $\mu_\pi$. In the Markovian setting, the process starts from $\mu_0$ and evolves along a single trajectory.

With these sampling regimes in place, we now turn to policy evaluation for $\pi$ using TD methods. Given a sampled transition $(s_t, a_t, s_t')$ at iteration $t$, TD with one-step bootstrapping (referred to as TD(0)) updates the value estimate as follows:

$$V(s_t) \leftarrow V(s_t) + \eta_t \left( r(s_t) + \gamma V(s_t') - V(s_t) \right)$$
$$\text{(TD(0) update)}$$

where $\eta_t > 0$ is the step-size at time $t$. For simplicity, we omit the superscript $\pi$ in $V^\pi$.

**Linear value function approximation.** The above TD(0) update is done on a per-state basis, and becomes computationally expensive for MDPs with a large state space. Consequently, previous works (Tsitsiklis & Roy, 1997; Bhandari et al., 2018) consider a linear approximation of the value function. Specifically, for parameters $w \in \mathbb{R}^d$ to be estimated, these works assume that $V_w(s) = w^\top \phi(s)$, where $\phi(s) \in \mathbb{R}^d$ is the known feature vector of state $s$. Given a sampled transition $(s_t, a_t, s_t')$, the linear TD(0) update (Sutton, 1988) is given by:

$$w_{t+1} = w_t + \eta_t \left( r(s_t) + \gamma w_t^\top \phi(s_t') - w_t^\top \phi(s_t) \right) \phi(s_t)$$
$$= w_t + \eta_t g_t(w_t). \tag{1}$$

where $g_t(w_t) := \left( r(s_t) + \gamma w_t^\top \phi(s_t') - w_t^\top \phi(s_t) \right) \phi(s_t)$ is the TD(0) direction at iteration $t$ and $\eta_t$ is the corresponding

step-size. It is convenient to define the expected TD(0) update direction $g(w)$ where the expectation is over the stationary distribution $\mu_\pi$. In particular, $g(w) := \mathbb{E}_{s \sim \mu_\pi, s' \sim P(\cdot|s)} \left[ \phi(s) \left( r(s) + (\gamma\phi(s') - \phi(s))^\top w \right) \right]$, referred to as the *mean-path update*. To analyze convergence, we adopt the following standard MDP assumptions:

**Assumption 2.1.** The Markov chain induced by policy $\pi$ is irreducible and aperiodic, and there exists a unique stationary distribution $\mu_\pi$.

The next assumption concerns the feature vectors used in the linear approximation. Let $n$ denote the number of states. Define $\Phi \in \mathbb{R}^{n \times d}$ as the feature matrix whose $i$-th row is $\phi(s_i)^\top$, and $D := \mathrm{diag}(\mu_\pi(s_1), \dots, \mu_\pi(s_n))$ as the diagonal matrix of stationary state probabilities. Finally, let $\Sigma := \Phi^\top D \Phi$, and let $\omega$ denote its smallest eigenvalue.

**Assumption 2.2.** The feature matrix $\Phi = [\phi(s_1)^\top, \dots, \phi(s_n)^\top] \in \mathbb{R}^{n \times d}$ has full column rank, which ensures a unique solution $w^*$. In addition, $\|\phi(s)\|^2 \leq 1$ for all $s$.

Under Assumption 2.1 and Assumption 2.2, TD(0) with suitable step-size $\eta_t$ converges to the unique fixed point $w^*$ with $g(w^*) = 0$ (Bhandari et al., 2018). We next study how to choose $\eta_t$ for the TD(0) update in Equation (1).

**Exponential step-size schedule.** We adopt an exponential schedule (Li et al., 2021). For a fixed number of iterations $T$, set $\eta_t = \eta_0 \, \alpha^t$ with $\alpha = (1/T)^{1/T}$. This schedule is effective for smooth, strongly convex problems and adapts to noise without prior knowledge of its level.

## 3. Exponential step-size with i.i.d. sampling

Prior analyses under i.i.d. sampling either set step sizes based on problem-dependent constants (Bhandari et al., 2018; Mustafin et al., 2024) or provide guarantees only for an averaged iterate (Bhandari et al., 2018; Samsonov et al., 2024), limiting practical utility. We adopt an exponential step-size schedule and develop a variant of TD(0) for i.i.d. sampling that does not require problem-dependent constants and establishes a last-iterate guarantee. We first present optimization-style lemmas that we will use, and then show how the exponential schedule delivers our objective.

We first provide a one-step expansion as follows.

$$\mathbb{E}\left[\|w_{t+1} - w^*\|^2\right] = \|w_t - w^*\|^2 \\ + \underbrace{2\eta_t \mathbb{E}_{s_t \sim \mu_\pi} \left[g_t(w_t)^\top (w_t - w^*)\right]}_{\text{progress}} \\ + \underbrace{\eta_t^2 \mathbb{E}_{s_t \sim \mu_\pi} \left[\|g_t(w_t)\|^2\right]}_{\text{variance}}.$$

The expectation is taken over i.i.d. sampling from the stationary distribution $\mu_\pi$. We omit the subscript below for

brevity. We next provide lemmas that give upper bounds on the progress and variance terms. The progress term can be analyzed using the following lemmas.

**Lemma 3.1.** *[Lemma 3 from Bhandari et al. (2018)] Under the i.i.d. sampling, $\forall w, w^* \in \mathbb{R}^d$,*

$$\mathbb{E}\left[g_t(w_t)^\top (w_t - w^*)\right] \geq (1 - \gamma) \|V_w - V_{w^*}\|_D^2,$$

This is analogous to a one-point strong monotonicity (or strong convexity) condition in optimization analysis. It lower bounds the alignment between the direction from the optimum to the current iterate $w^* - w_t$, and the update direction $g_t(w)$ by the value-space error $\|V_w - V_{w^*}\|_D^2$.

**Lemma 3.2.** *[Extended Lemma 1 from Bhandari et al. (2018)] Under the i.i.d. sampling, $\forall w_1, w_2 \in \mathbb{R}^d$,*

$$\omega \|w_1 - w_2\|^2 \leq \|V_{w_1} - V_{w_2}\|_D^2 = \|w_1 - w_2\|_\Sigma^2$$

This lemma lower bounds $\|V_{w_1} - V_{w_2}\|_D^2$ with $\omega \|w_1 - w_2\|^2$, allowing us to use strong-convexity-style arguments. Using these lemmas, we can bound the progress term as follows:

$$2\eta_t \mathbb{E}\left[g_t(w_t)^\top (w_t - w^*)\right] \leq -2\eta_t (1 - \gamma) \omega \|w_t - w^*\|^2.$$

In order to bound the variance term, we use the following lemma.

**Lemma 3.3.** *[Lemma 5 from Bhandari et al. (2018)] Under the i.i.d. sampling, $\mathbb{E}\left[\|g_t(w)\|^2\right] \leq 2\sigma^2 + 8 \|V_w - V_{w^*}\|_D^2$, where $\sigma^2 = \mathbb{E}\left[\|g_t(w^*)\|^2\right]$.*

$\sigma^2$ is the variance of the TD update at the optimum. This lemma is analogous to the variance control in optimization.

Combining these bounds on the progress and variance terms and setting $\eta_0 \leq \frac{1-\gamma}{8}$, we get the following bound:

$$\mathbb{E}\left[\|w_{t+1} - w^*\|^2\right] \\ \leq \|w_t - w^*\|^2 \left(1 - 2\eta_0 \alpha^t (1 - \gamma)\omega\right) + 2\eta_0^2 \alpha^{2t} \sigma^2.$$

Taking expectation over $t \in [T]$ and using the fact $(1-x) \leq e^{-x}$, we have

$$\mathbb{E}\left[\|w_T - w^*\|^2\right] \\ \leq \|w_0 - w^*\|^2 \exp\left(-\eta_0 \omega(1-\gamma) \sum_{t=1}^T \alpha^t\right) \\ + 2\sigma^2 \eta_0^2 \sum_{t=1}^T \alpha^{2t} \exp\left(-\eta_0 \omega(1-\gamma) \sum_{i=t+1}^T \alpha^i\right).$$

As shown in Appendix F, the exponential step-size yields $\sum_{t=1}^T \alpha^t \geq \frac{\alpha T}{\ln T} - \frac{1}{\ln T}$ and

$\sum_{t=1}^{T} \alpha^{2t} \exp\left(-\sum_{i=t+1}^{T} \alpha^i\right) \leq O\left(\frac{(\ln(T))^2}{\alpha^2 T}\right)$. The exponential step-size achieves bias-variance trade-off without iterate averaging (Patil et al., 2023; Samsonov et al., 2024). Combining these bounds gives the final rate:

**Theorem 3.4.** *Under Assumption 2.1 and 2.2, TD(0) under i.i.d. sampling from the stationary distribution with $\eta_t = \eta_0 \alpha_t$, where $\eta_0 = \frac{1-\gamma}{8}$, $\alpha_t = \alpha^t = \frac{1}{T}^{t/T}$, $\alpha = \frac{1}{T}^{1/T}$, has the following convergence:*

$$\mathbb{E}\left[\|w_{T+1} - w^*\|^2\right]$$

$$\leq \|w_1 - w^*\|^2 e \exp\left(-\eta_0 \omega (1-\gamma) \frac{\alpha T}{\ln T}\right)$$

$$+ \frac{8\sigma^2}{e\left(\omega(1-\gamma)\right)^2} \frac{\ln^2 T}{\alpha^2 T},$$

*where $\sigma^2 = \mathbb{E}\left[\|g_t(w^*)\|^2\right]$.*

The proof of this theorem is in Appendix D. Our analysis adopts the optimization perspective in Bhandari et al. (2018). Compared with other methods in Table 1, our main contribution here is showing how exponential step-sizes, combined with optimization-based arguments, yield a cleaner proof and improved convergence guarantees for the last iterate. *We emphasize that for TD(0) in the i.i.d. setting, ours is the first work that attains the optimal bias-variance trade-off for the last iterate.*

Compared to the most relevant prior works, we note that Bhandari et al. (2018) either obtain a slower $O(1/\sqrt{T})$ rate or require knowledge of $\omega$ to obtain faster rates without optimal trade-off between the bias and variance. In contrast, the exponential step-size attains the optimal bias-variance trade-off without requiring difficult-to-estimate problem-dependent quantities. Samsonov et al. (2024); Patil et al. (2023) establish a similar convergence bound attaining the optimal bias-variance trade-off with a universal step-size that does not rely on unknown problem-dependent constants. Compared to Theorem 3.4, both these works rely on the stochastic approximation perspective, and derive the convergence rate only for the averaged iterate. Compared to Samsonov et al. (2024); Patil et al. (2023), Theorem 3.4 has an additional $O(\ln(T))$ dependence in the variance term. This additional log factor comes from Lemmas F.1 and F.2 and also appears in stochastic smooth, strongly convex minimization (see Vaswani et al. (2022)). Hence, our results provide a complementary bias–variance trade-off: we accept a mild log factor in exchange for a meaningful last-iterate guarantee without averaging. In the future, we will investigate whether these bounds can be sharpened to reduce or remove the extra log factors.

One limitation of our result is the quadratic dependence on $\omega$. With respect to the iteration count, our $\tilde{O}(1/T)$ variance term matches the standard $\Theta(1/T)$ lower bound for smooth,

strongly convex stochastic minimization with diminishing step-sizes (Nguyen et al., 2018). With respect to $\omega$, all known parameter-free upper bounds, Patil et al. (2023), Samsonov et al. (2024) and ours, incur quadratic $O(\sigma^2/(\omega^2 T))$ dependence in the variance term, whereas methods that use $\omega$ to set the step-size achieve linear $O(\sigma^2/(\omega T))$ dependence (Bhandari et al., 2018). This consistent gap across all known upper bounds suggests an inherent price of adaptivity to unknown $\omega$, rather than a weakness specific to our approach. However, a formal lower bound establishing this separation specifically for parameter-free TD remains an open problem.

# 4. Handling Markovian sampling

We now relax the i.i.d. assumption and consider Markovian sampling, where the TD update uses samples drawn sequentially from a single trajectory of the Markov chain. This setting is more realistic because it does not assume that samples are drawn from the hard-to-estimate stationary distribution, but are instead collected by interacting with the environment. However, since the samples are temporally correlated, the update direction is biased relative to the mean-path update. In particular, when the chain is not at stationarity, in general $\mathbb{E}[g_t(w_t)] \neq g(w_t)$. This requires controlling an additional error term.

In order to do so, we will use the property that the Markov chain is fast-mixing. This is a standard assumption in the analysis of the TD algorithm (Bhandari et al., 2018; Mitra, 2025). In particular, under Assumption 2.1, the $t$-step state distribution $\mu_0 P_\pi^t$ started from any $\mu_0$ converges to the stationary distribution $\mu_\pi$ geometrically fast, *i.e.*,

$$\sup_{\mu_0} d_{\text{TV}}\left(P_\pi^t \mu_0, \mu_\pi\right) \leq m\rho^t, \forall t \in \mathbb{N}_0, \qquad (2)$$

where $d_{\text{TV}}$ is the total variation distance, and initial distance $m$ and mixing speed $\rho \in (0,1)$ are positive constants that depend on the underlying Markov chain. This deviation from stationarity is quantified via the *mixing time*.

**Definition 4.1.** Define the mixing time as $\tau_\delta = \min\{t \in \mathbb{N}_0 \mid m\rho^t \leq \delta\}$, where $\delta \in (0,1)$.

We define $\tau_{\text{mix}}$ as $\tau_\delta$ for an appropriate $\delta$ to be determined later.

Using this property of fast-mixing, Bhandari et al. (2018); Mitra (2025) controlled the error term from Markovian sampling. In particular, the analysis in Bhandari et al. (2018) requires projecting the iterates onto a bounded set containing $w^*$. This projection step is nonstandard in practice, and requires knowledge of $\omega$. Mitra (2025) avoids this projection step by using an induction argument to show the iterates remain bounded. However, they can only prove convergence for the average iterate (obtained by Polyak–Ruppert averaging) and require knowledge of both $\tau_{\text{mix}}$ and $\omega$. Unlike the

analysis in Mitra (2025), we show that using exponential step-sizes allows us to prove convergence for the *last iterate* without knowledge of $\tau_{\text{mix}}$. Moreover, to remove dependence on $\omega$, we use the regularized TD(0) update in Patil et al. (2023). The regularized TD(0) update at iteration $t$ is given by

$$w_{t+1} = w_t + \eta_t g_t^r(w), \quad \text{where}$$
$$g_t^r(w) := \phi(s_t)\left(r(s_t) + (\gamma\phi(s_{t+1}) - \phi(s_t))^\top w\right) - \lambda w$$
$$= g_t(w) - \lambda w,$$

and $\lambda > 0$ is the strength of regularization. The corresponding mean-path regularized direction is defined as

$$g^r(w) := \mathbb{E}\left[\phi(s_t)\left(r(s_t) + (\gamma\phi(s_{t+1}) - \phi(s_t))^\top w\right)\right] - \lambda w$$
$$= g(w) - \lambda w.$$

We define $w_r^*$ as the fixed point of the regularized TD(0) update satisfying $g^r(w_r^*) = 0$. Note that the standard TD(0) update involving $g_t(w)$ is a special case of $g_t^r(w)$ with $\lambda = 0$. We now establish the properties shared by both the regularized and standard TD(0) variants.

In particular, we first show that the following two lemmas provide upper bounds on $\|g_t^r(w)\|$ and $\|g^r(w)\|$ for the regularized and standard TD(0) variants.

**Lemma 4.2.** *For stochastic update $g_t^r$, we have*

$$\|g_t^r(w)\| \leq (2 + \lambda)\|w - w_r^*\| + (3 + \lambda)\zeta,$$

*where $\zeta = \max\{1, \|w_r^*\|\}$.*

*For standard TD(0) corresponding to $\lambda = 0$, $\|g_t^r(w)\| \leq 2\|w - w_r^*\| + 3\zeta$, $\zeta = \max\{1, \|w^*\|\}$.*

**Lemma 4.3.** *For mean-path update $g^r$, we have*

$$\|g^r(w)\| \leq (2 + \lambda)\|w - w_r^*\|,$$

*where $\zeta = \max\{1, \|w_r^*\|\}$. For standard TD(0) corresponding to $\lambda = 0$, $\|g^r(w)\| \leq 2\|w - w_r^*\|$.*

For Markovian sampling, we show that the fast-mixing property in Equation (2) implies the following result.

**Lemma 4.4.** *For any initial state distribution $\mu_0$, the state distribution at time $t$ is $P_\pi^t \mu_0$. For any $w$, when $t \geq \tau_\delta$,*

$$\left\|\mathbb{E}_{s_t \sim P_\pi^t \mu_0}[g_t^r(w)] - g^r(w)\right\| \leq 2(2 + \lambda)\delta(\|w\| + 1).$$

*For standard TD(0) corresponding to setting $\lambda = 0$,*

$$\left\|\mathbb{E}_{s_t \sim P_\pi^t \mu_0}[g_t(w)] - g(w)\right\| \leq 4\delta\|w\| + 1.$$

For the initial step-size $\eta_0$ in the TD(0) update, for a fixed $T$, we set $\delta$ and define $\tau_{\text{mix}}$ as follows:

$$\delta = \frac{\eta_0}{2(2 + \lambda)T} \quad ; \quad \tau_{\text{mix}} = \tau_\delta. \tag{3}$$

By Equation (2), when $T \geq \frac{\ln(2(2+\lambda)Tm/\eta_0)}{\ln(1/\rho)}$, this implies that $T \geq \tau_{\text{mix}} := \frac{\ln(2(2+\lambda)Tm/\eta_0)}{\ln(1/\rho)}$. Furthermore, using Lemma 4.4 implies that the following property holds for all $t \geq \tau_{\text{mix}}$,

$$\left\|\mathbb{E}_{s_t \sim P_\pi^t \mu_0}[g_t^r(w)] - g^r(w)\right\| \leq \eta_0 \frac{1}{T}(\|w\| + 1)$$
$$= \eta_T(\|w\| + 1), \tag{4}$$

where $\eta_0$ and $\eta_T$ are the step-sizes at iterations $t = 0$ and $t = T$ in the exponential step-size schedule $\eta_t = \eta_0 \alpha^t$, respectively. The above equation shows that the deviation between the expected update direction at iteration $t$, $g_t(w)$, and the corresponding mean-path update $g(w)$ becomes small once $t \geq \tau_{\text{mix}}$.

In the next step, we bound one-step progress similar to the i.i.d. case in Section 3. In particular, we separate the Markovian component $g_t^r(w_t) - g^r(w_t)$ from the corresponding mean-path term, and use the following bound.

$$\|w_{t+1} - w_r^*\|^2$$
$$\overset{(i)}{=} \|w_t - w_r^*\|^2 + 2\eta_t\langle g_t^r(w_t), w_t - w_r^*\rangle + \eta_t^2\|g_t^r(w_t)\|^2$$
$$= \|w_t - w_r^*\|^2 + 2\eta_t\langle g_t^r(w_t) - g^r(w_t), w_t - w_r^*\rangle$$
$$\quad + \eta_t^2\|g_t^r(w_t)\|^2 + 2\eta_t\langle g^r(w_t), w_t - w_r^*\rangle$$
$$\implies \mathbb{E}_{s_t \sim P_\pi^t \mu_0}\left[\|w_{t+1} - w_r^*\|^2\right]$$
$$\overset{(ii)}{\leq} \underbrace{\|w_t - w_r^*\|^2 + 2\eta_t\langle g^r(w_t), w_t - w_r^*\rangle + 2\eta_t^2\|g^r(w_t)\|^2}_{\text{mean-path}}$$
$$+ \underbrace{2\eta_t\mathbb{E}_{s_t \sim P_\pi^t \mu_0}\left[\langle g_t^r(w_t) - g^r(w_t), w_t - w_r^*\rangle\right]}_{\text{Markovian drift}}$$
$$+ \underbrace{2\eta_t^2\mathbb{E}_{s_t \sim P_\pi^t \mu_0}\left[\|g_t^r(w_t) - g^r(w_t)\|^2\right]}_{\text{Markovian variance}}, \tag{5}$$

where (i) uses the TD(0) update and (ii) uses the fact that $\|a\|^2 \leq 2\|a - b\|^2 + 2\|b\|^2$. Note that the two terms involving $g_t^r(w_t) - g^r(w_t)$, Markovian drift and Markovian variance, capture the Markovian noise, while the remaining mean-path term only depends on mean-path quantities. Unlike the i.i.d. setting, since $g_t$ and $w_t$ are correlated even after conditioning on the randomness at iteration $t$, $\mathbb{E}_{s_t \sim P_\pi^t \mu_0}\left[\langle g_t^r(w_t) - g^r(w_t), w_t - w_r^*\rangle|w_t\right] \neq 0$. Consequently, we follow the proof in Mitra (2025) and use a strong induction argument to simultaneously control the Markovian drift and Markovian variance terms and show that the iterates remain bounded, *i.e.*, for a constant $B(\tau_{\text{mix}})$ that depends on the mixing time, $\|w_t - w_r^*\|^2 \leq B(\tau_{\text{mix}})$.

For this, we first note that for $t \geq \tau_{\text{mix}}$, Equation (4) allows us to bound the average discrepancy between $g_t^r(w)$ and $g^r(w)$ in terms of $\|w\|$. Hence, in order to set up the induction we use $t = \tau_{\text{mix}}$ as the base case and first show that $\|w_t - w_r^*\|$ is bounded by $B(\tau_{\text{mix}})$ for *all* $t \leq \tau_{\text{mix}}$.

**Lemma 4.5.** *For the regularized TD(0) update with exponential step-sizes $\eta_t = \eta_0 \alpha_t$, where $\eta_0 \le \frac{1-\gamma}{16 \ln(T)}$, $\alpha_t = \alpha^t = \frac{1}{T}^{t/T}$, $\alpha = \frac{1}{T}^{1/T}$, if $T \ge \max\{3, 1/\eta_0\}$,*

$$\forall t \le \tau_{mix}, \quad \|w_t - w_r^*\|^2 \le B(\tau_{mix}) \quad \textbf{(Base case)},$$

*where $B(\tau_{mix}) := \exp(2(2+\lambda)\max\{a,b\}) \cdot \|w_1 - w_r^* + \zeta\|^2$, where $a = \frac{1}{\ln(1/\rho)}$, $b = \frac{\ln(2(2+\lambda)m)}{\ln(1/\rho)}$, $\zeta = \max\{1, \|w_r^*\|\}$.*

With Lemma 4.5 giving the bound on $\|w_t - w_r^*\|^2$ for iterations $t \le \tau_{\text{mix}}$, the base case for the induction is set up. We now state the inductive hypothesis for iteration $t$.

**Inductive Hypothesis**: For a fixed $t$, for all $k \le t$, $\|w_k - w_r^*\|^2 \le B(\tau_{\text{mix}})$.

**Inductive Step**: To complete the induction, we need to show that for a fixed $t$, for all $k \le t+1$, $\|w_k - w_r^*\|^2 \le B(\tau_{\text{mix}})$. In order to do so, we use the inductive hypothesis and first prove a lemma that controls the size of the update across $\tau_{\text{mix}}$ iterations. Specifically, for $t \ge \tau_{\text{mix}}$, we bound $w_t$ in terms of $w_{t-\tau_{\text{mix}}}$ as follows.

**Lemma 4.6.** *Let $T \ge \max\{3, \frac{1}{\eta_0}\}$, and let $T$ be large enough that $\frac{\ln^2(T)}{T} \le \frac{1}{2a}$, $\frac{\ln(T)}{T} \le \frac{1}{b}$, and $\ln(T) \ge \max\{a, b\}$. Suppose for all $t \ge \tau_{mix}$, if $\|w_k - w_r^*\|^2 \le B(\tau_{mix})$ for all $k \in [t]$, then,*

$$\|w_t - w_{t-\tau_{mix}}\|^2 \le c_1^2 \, B(\tau_{mix}) \, \eta_t^2 \, \ln^4(T),$$

*where $c_1^2 = 2560(2+\lambda)^2$, $a = \frac{1}{\ln(1/\rho)}$ and $b = \frac{\ln(2(2+\lambda)m)}{\ln(1/\rho)}$.*

Next, we decompose the Markovian drift term for all $t \ge \tau_{\text{mix}}$. In particular,

$$\mathbb{E}_t[\langle g_t^r(w_t) - g^r(w_t), w_t - w_r^*, \rangle] = T_1 + T_2 + T_3 + T_4$$

s.t $T_1 = \mathbb{E}_t[\langle w_t - w_{t-\tau_{\text{mix}}}, g_t^r(w_t) - g^r(w_t)\rangle]$,
$T_2 = \mathbb{E}_t[\langle w_{t-\tau_{\text{mix}}} - w_r^*, g_t^r(w_{t-\tau_{\text{mix}}}) - g^r(w_{t-\tau_{\text{mix}}})\rangle]$,
$T_3 = \mathbb{E}_t[\langle w_{t-\tau_{\text{mix}}} - w_r^*, g_t^r(w_t) - g_t^r(w_{t-\tau_{\text{mix}}})\rangle]$,
$T_4 = \mathbb{E}_t[\langle w_{t-\tau_{\text{mix}}} - w_r^*, g^r(w_{t-\tau_{\text{mix}}}) - g^r(w_t)\rangle]$.

Note that terms $T_1$, $T_3$ and $T_4$ can be bounded deterministically by using Young's inequality. In particular, $T_1$ can be bounded using Lemma 4.6 and the uniform bound on $\|g_t^r(w)\|$ and $\|g^r(w)\|$ given by Lemma 4.2 and Lemma 4.3. Terms $T_3$ and $T_4$ can be bounded by using the Lipschitzness of $g^r$, Lemma 4.6 and using the inductive hypothesis to bound $\|w_{t-\tau_{\text{mix}}} - w_r^*\|$. For $T_2$, we use the bound from Equation (4) after conditioning on $w_{t-\tau_{\text{mix}}}$. Summing these four terms yields the following lemma:

**Lemma 4.7.** *Let $T \ge \max\{3, \frac{1}{\eta_0}\}$, and let $T$ be large enough that $\frac{\ln^2(T)}{T} \le \frac{1}{2a}$, $\frac{\ln(T)}{T} \le \frac{1}{b}$, and $\ln(T) \ge$*

$\max\{a, b\}$. *For $t \ge \tau_{mix}$, suppose $\|w_k - w_r^*\|^2 \le B(\tau_{mix})$ for all $k \in [t]$. Then:*

$$\mathbb{E}_t \left[\langle g_t^r(w_t) - g^r(w_t), \, w_t - w_r^* \rangle\right]$$
$$\le C \, \eta_t \, \ln^2(T) \, B(\tau_{mix}),$$

*where $C = C_1 + 3 + 2C_2$, $C_1 = \frac{c_1}{2}$ and $C_2 = \frac{c_1 c_2}{2}$, $c_1 = 2560(2+\lambda)^2$ and $c_2 = 4(2+\lambda)^2 + 4(3+\lambda)^2 + 2(2+\lambda)^2$.*

Finally, using the inductive hypothesis and the uniform bound on $\|g_t^r(w)\|$ and $\|g^r(w)\|$, we can show that the Markovian variance term in Equation (5) can be deterministically bounded in terms of $B(\tau_{\text{mix}})$. The following lemma provides this bound.

**Lemma 4.8.** *Assuming $\|w_k - w_r^*\|^2 \le B(\tau_{mix}), \forall k \in [t]$, then we have*

$$\mathbb{E}_{s_t \sim P_\pi^t \mu_0} \left[\|g_t^r(w_t) - g^r(w_t)\|^2\right] \le C' B(\tau_{mix}),$$

*where $C' = 10(3+\lambda)^2$.*

In order to complete the inductive step, we use Equation (5) and the bounds on the Markovian drift and Markovian variance terms along with the mean-path analysis for the mean-path term to show that $\|w_{t+1} - w_r^*\|^2 \le B(\tau_{\text{mix}})$ and therefore for all $k \le t+1$, $\|w_k - w_r^*\|^2 \le B(\tau_{\text{mix}})$. For this last step, the analysis for the standard and regularized TD(0) updates is different, and we handle them separately.

### 4.1. Standard TD(0)

We use the above lemmas with $\lambda = 0$. Consequently, $w_r^* = w^*$ and $g^r(w) = g(w)$ where $g(w^*) = 0$. The following lemma shows that if the initial step-size $\eta_0$ is small enough, then we can use the inductive hypothesis to show that $\|w_{t+1} - w_r^*\|^2 \le B(\tau_{\text{mix}})$.

**Lemma 4.9.** *For the standard TD(0) update, when $T \ge \max\{3, \frac{1}{\eta_0}, \frac{\ln(4Tm/\eta_0)}{\ln(1/\rho)}\}$, and $T$ is large enough that $\frac{\ln^2(T)}{T} \le \frac{1}{2a}$, $\frac{\ln(T)}{T} \le \frac{1}{b}$, and $\ln(T) \ge \max\{a, b\}$, for a fixed $t$, if $\|w_k - w_r^*\|^2 \le B(\tau_{mix})$ for all $k \le t$ and*

$$\eta_0 \le \frac{(1-\gamma)\omega}{2[C \ln^2(T) + C']},$$

*then $\|w_{t+1} - w^*\|^2 \le B(\tau_{mix})$, and hence, $\|w_k - w_r^*\|^2 \le B(\tau_{mix})$ for all $k \le t+1$.*

This completes the induction for standard TD(0), and shows that for all $t \in [T]$, $\|w_t - w^*\|^2 \le B(\tau_{\text{mix}})$. Using Lemma 4.7, this also implies that the Markovian drift term in Equation (5) is bounded for all $t \in [T]$. Similarly, the Markovian variance term is also bounded for all $t \in [T]$ as shown in Lemma 4.8. Putting together these results, we state the complete theorem for the standard TD(0) update.

**Theorem 4.10.** *The standard TD(0) update with exponential step-sizes $\eta_t = \eta_0 \alpha_t$, where $\eta_0 = \frac{(1-\gamma)\omega}{2[C \ln^2(T) + C']}$, $\alpha_t = \alpha^t = \frac{1}{T}^{t/T}$, and $T \geq \max\{\frac{1}{\eta_0}, \frac{\ln(4Tm/\eta_0)}{\ln(1/\rho)}\}$, and $T$ is large enough that $\frac{\ln^2(T)}{T} \leq \frac{1}{2a}$, $\frac{\ln(T)}{T} \leq \frac{1}{b}$, and $\ln(T) \geq \max\{a, b\}$, achieves the following convergence rate:*

$$\mathbb{E}\left[\|w_{T+1} - w^*\|^2\right]$$
$$= O\left(\exp\left(-\frac{\omega^2 T}{\ln^3(T)}\right) + \frac{\ln^4(T)}{\omega^2 T} \exp\left(\frac{m}{\ln(1/\rho)}\right)\right),$$

*where $m$ and $\rho$ are related to mixing time as $\tau_{mix} = \frac{\ln(4Tm/\eta_0)}{\ln(1/\rho)}$.*

The complete proof can be found in Appendix E. Compared with other methods in Table 1, standard TD(0) with exponential step-size achieves a fast convergence rate without requiring projection onto a bounded set. In addition, our guarantee is for the last iterate. Compared with our i.i.d. sampling result in Section 3, the rate under Markovian sampling is comparable. However, it requires a problem-dependent parameter $\omega$ to set the initial step-size $\eta_0$. In the next subsection, we will show that the regularized TD(0) update removes the dependence on $\omega$.

## 4.2. Regularized TD(0)

In this section, we analyze regularized TD(0). The following lemma provides a step-size condition that, under the inductive hypothesis, guarantees $\|w_t - w_r^*\|^2 \leq B(\tau_{mix})$ for all $t \in [T]$.

**Lemma 4.11.** *For the regularized TD(0) update, when $T \geq \max\{3, \frac{1}{\eta_0}, \frac{\ln(2(2+\lambda)Tm/\eta_0)}{\ln(1/\rho)}\}$, and $T$ is large enough that $\frac{\ln^2(T)}{T} \leq \frac{1}{2a}$, $\frac{\ln(T)}{T} \leq \frac{1}{b}$, and $\ln(T) \geq \max\{a, b\}$, for a fixed $t$, if $\|w_k - w_r^*\|^2 \leq B(\tau_{mix})$ for all $k \leq t$ and the step-size satisfies*

$$\eta_0 \leq \frac{\lambda}{[C \ln^2(T) + C'] + (8 + 2\lambda^2)},$$

*then $\|w_{t+1} - w_r^*\|^2 \leq B(\tau_{mix})$, and hence, $\|w_k - w_r^*\|^2 \leq B(\tau_{mix})$ for all $k \leq t + 1$.*

This completes the induction for regularized TD(0). Compared with the step-size requirement in Lemma 4.9, regularized TD(0) removes the requirement for $\omega$ and replaces it with $\lambda$, a regularization parameter that can be set appropriately. As in standard TD(0), with Lemma 4.7 and Lemma 4.8, we obtain upper bounds on the Markovian drift and Markovian variance terms for $t \in [T]$. Having bounded all terms in the one-step expansion for all $t \in [T]$, we combine these bounds and state the final convergence rate, also accounting for the distance between $w_r^*$ and $w^*$.

**Theorem 4.12.** *The regularized TD(0) update with exponential step-size $\eta_t = \eta_0 \alpha_t$, where $\eta_0 = \frac{\lambda}{[C \ln^2(T) + C'] + (8 + 2\lambda^2)}$, $\alpha_t = \alpha^t = \frac{1}{T}^{t/T}$, and $T \geq \max\{\frac{1}{\eta_0}, \frac{\ln(2(2+\lambda)Tm/\eta_0)}{\ln(1/\rho)}\}$, and $T$ is large enough that $\frac{\ln^2(T)}{T} \leq \frac{1}{2a}$, $\frac{\ln(T)}{T} \leq \frac{1}{b}$, $\ln(T) \geq \max\{a, b\}$, and $\lambda = 1/\sqrt{T}$, achieves the following convergence rate:*

$$\mathbb{E}\left[\|w_{T+1} - w^*\|^2\right]$$
$$= O\left(\exp\left(-\frac{\omega\sqrt{T}}{\ln^3(T)}\right) + \frac{\ln^4(T)}{\omega^2 T} \exp\left(\frac{m}{\ln(1/\rho)}\right)\right),$$

*where $m$ and $\rho$ are related to mixing time as $\tau_{mix} = \frac{\ln(2(2+\lambda)Tm/\eta_0)}{\ln(1/\rho)}$.*

**Technical novelty compared to Mitra (2025).** Our proof builds on the i.i.d. setting from Section 3 and the induction technique of Mitra (2025). Compared to Mitra (2025), our key technical innovations are threefold. (a) In the proofs of Lemmas 4.5 to 4.7, Lemma E.3 through Lemma E.6, we establish key connections between exponential step-sizes and Markovian sampling quantities, in particular, the mixing parameters $(m, \rho)$. These lemmas allow us to control the terms involving $\tau_{mix}$ and eliminate the need to set the step-size using $\tau_{mix}$. In contrast, Mitra (2025) set the step-size according to $\tau_{mix}$ (see Lemma 2 of Mitra (2025)). (b) Mitra (2025) derive convergence by splitting the analysis before and after the mixing time and selecting step sizes via an impractical comparison, specifically between $\frac{\ln(\lambda)}{0.5\omega(1-\gamma)(T+1)}$ and $\frac{\omega(1-\gamma)}{C\tau_{mix}}$ (see the proof of Theorem 3). In contrast, our analysis yields last-iterate convergence without such case distinctions or comparisons, and without relying on unknown constants. (c) Our use of regularization to remove the dependence on $\omega$ is novel in this line of work; in comparison, Mitra (2025) require knowledge of $\omega$ to set algorithm parameters (see Theorem 1, Theorem 3, and the proof in their appendix).

**Comparison to Haque et al. (2024).** Haque et al. (2024) also shows the convergence of linear TD without requiring knowledge of problem-dependent constants for setting the step-size. While Haque et al. (2024) attain the desired convergence for the average iterate, we use an additional regularization to attain the more practical last-iterate convergence. While both works derive an $O(1/T)$ rate, our result has an additional $O(\ln(T))$ term in the leading term. On the other hand, we have an explicit dependence on the fast-decaying transient term and the mixing time, whereas this dependence is not clear for Haque et al. (2024). Finally, we note that the analysis in Haque et al. (2024) uses tools from stochastic approximation, whereas our proof uses standard convex optimization techniques.

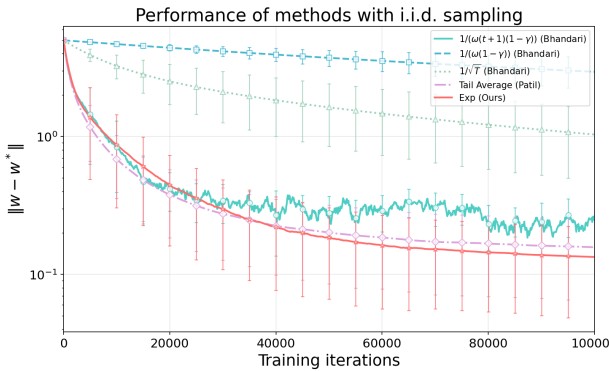

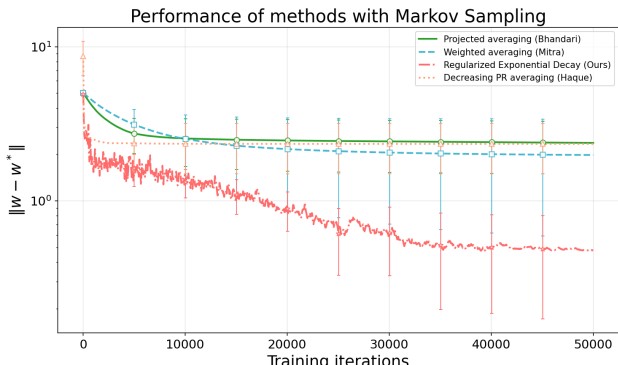

*Figure 1.* Convergence under **i.i.d. sampling**: mean $\|w_t - w^*\|$ over 10 seeds ($\pm 1$ std shaded). The exponential schedule (Ours) is parameter-free and last-iterate.

*Figure 2.* Convergence under **Markovian sampling** ($\lambda_{\min}(\Phi\Phi^\top) = 10^{-4}$, SLEM $\approx 8 \times 10^{-4}$); step-sizes swept per method and the best reported.

Compared to the methods in Table 1, Bhandari et al. (2018) do not obtain optimal trade-off between the bias and the mixing time, and require projecting onto a ball, which is nonstandard and requires knowledge of $\omega$. Patil et al. (2023) analyze the Markovian case with data drop, an impractical algorithm that also requires knowledge of $\tau_{\text{mix}}$. Our method requires no projection onto a bounded set or any prior knowledge of problem-dependent parameters. In addition, our guarantee is for the last iterate, which is often more practical than iterate averaging.

Compared to previous works, one limitation of Theorem 4.12 is the extra term $\exp\left(\frac{m}{\log(1/\rho)}\right) \approx \exp\left(\frac{m}{1-\rho}\right) \approx \exp(m\tau)$, which induces an exponential dependence on the mixing time. This dependence is therefore weaker than the linear dependence in prior works. We conjecture that this worse dependence is an artifact of our analysis, and improving this is an important direction for future work. The second limitation of Theorem 4.12 is that our step-size schedule assumes knowledge of the horizon $T$. We use the doubling trick (Appendix G) to remove this requirement, while preserving the original convergence rate.

## 5. Experiments

We validate the exponential schedule on a synthetic MDP that lets us systematically control $\omega$ and the mixing time $\tau_{\text{mix}}$ with zero approximation error; the full construction and per-method settings are in Appendix H. Figures 1 and 2 show the results under i.i.d. and Markovian sampling, respectively.

## 6. Conclusion

We address the sensitivity of TD learning to step-size selection and unknown problem parameters by using an exponential schedule $\eta_t = (1/T)^{t/T}$. Our main contributions are twofold. First, under both i.i.d. and Markovian sam-

pling, our method requires no prior knowledge of problem-dependent constants and, in the Markovian case, avoids projections. Second, we prove finite-time, last-iterate convergence guarantees in both settings. Overall, these results suggest a more practical alternative for TD learning with reduced step-size tuning. The exponential schedule and regularization are *sufficient*, not necessary, for parameter-free last-iterate TD. AdaGrad-style step-sizes, for instance, attain parameter-free $O(1/\sqrt{T})$ rates in the convex setting; whether such alternatives extend to TD, and whether un-regularized parameter-free Markovian TD is achievable, remain open. For future work, we view high-probability guarantees as an important direction. A formal extension to infinite-dimensional kernel TD also appears tractable: under standard polynomial or exponential decay of the kernel operator's spectrum, the algorithm makes meaningful progress along directions with eigenvalue $\lambda_i \geq \Omega(\text{polylog}(T)/T)$, and the effective dimension is governed by this spectral threshold.

## Acknowledgments

We would like to thank Wenlong Mou, Qiushi Lin and Xingtu Liu for helpful feedback on the paper. This work was partially supported by the Canada CIFAR AI Chair Program, the Natural Sciences and Engineering Research Council of Canada (NSERC) Discovery Grants RGPIN-2022-03669, and enabled in part by support provided by the Digital Research Alliance of Canada (alliancecan.ca).

## Impact Statement

This paper presents work whose goal is to advance the field of Machine Learning. There are many potential societal consequences of our work, none of which we feel must be specifically highlighted here.

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

# A. Additional Inequalities Used in the Proofs

There are other inequalities regarding $g(w)$ that we used in the proofs. We list them here for completeness.

$-g$ **is** $2$**-Lipschitz.**   *i.e.* $\|g(w_1) - g(w_2)\| \le 2\|w_1 - w_2\|$.

*Proof.*
$$\|g(w_1) - g(w_2)\| = \mathbb{E}_{s_t \sim \mu_\pi, s_{t+1} \sim P_\pi \mu_\pi} \left[ \left\| \phi(s_t)(\phi(s_t) - \gamma\phi(s_{t+1}))^\top (w_1 - w_2) \right\| \right] \tag{6}$$
$$\le 2\|w_1 - w_2\|.$$

$\square$

$-g_t$ **is** $2$**-Lipschitz.**   *i.e.* $\|g_t(w_1) - g_t(w_2)\| \le 2\|w_1 - w_2\|$.

*Proof.*
$$\|g_t(w_1) - g_t(w_2)\| = \left\| \left[ \phi(s_t)(\phi(s_t) - \gamma\phi(s_{t+1}))^\top \right] (w_1 - w_2) \right\| \tag{7}$$
$$\le 2\|w_1 - w_2\|.$$

$\square$

**Equations 7 and 8 in Mitra (2025).**
$$\|g_t(w)\| \le 2\|w - w^*\| + 4\zeta, \tag{8}$$
where $\zeta = \max\{1, \|w^*\|\}$. Since $g(w^*) = 0$, we have
$$\|g(w)\| \le 2\|w - w^*\|. \tag{9}$$

# B. Proof for Constant step-size with mean-path update

We follow the vanilla TD(0) update proof in Bhandari et al. (2018).

**Theorem B.1.** *For the mean-path TD(0) update with a constant step-size $\eta \le \frac{1-\gamma}{8}$, the algorithm achieves the following convergence rate:*
$$\|w_T - w^*\|^2 \le \exp\left(-\eta(1-\gamma)\omega T\right) \left[ \|w_1 - w^*\|_D^2 \right].$$

*Hence, to obtain accuracy $\|w_T - w^*\| \le \epsilon$, we need $O\left(\left(\frac{1}{\omega}\right) \log\left(\frac{1}{\epsilon}\right)\right)$ gradient evaluations.*

*Proof.* With the update $w_{t+1} = w_t + \eta g(w_t)$
$$\|w_{t+1} - w^*\|^2 = \|w_t - w^*\|^2 + 2\eta g(w_t)^\top (w_t - w^*) + \eta^2 \|g(w_t)\|^2.$$

$$
\begin{aligned}
\|w_{t+1} - w^*\|^2 &= \|w_t - w^*\|^2 + 2\eta g(w_t)^\top (w_t - w^*) + \eta^2 \|g(w_t)\|^2 \\
&\le \|w_t - w^*\|^2 - \left(2\eta(1-\gamma) - 8\eta^2\right) \|V_{w^*} - V_{w_t}\|_D^2 && \text{(by Lemma 3.1 Lemma 3.3)} \\
&\le \|w_t - w^*\|^2 - \eta(1-\gamma)\|V_{w^*} - V_{w_t}\|_D^2 && (\eta \le (1-\gamma)/8) \\
&\le \|w_t - w^*\|^2 - \eta(1-\gamma)\omega\|w_t - w^*\|^2. && \text{(by Lemma 3.2)} \\
&= (1 - \eta(1-\gamma)\omega)\|w_t - w^*\|^2
\end{aligned}
$$

Recursing over $t \in [T]$, we have
$$\|w_T - w^*\|^2 \le (1 - \eta(1-\gamma)\omega)^T \left[ \|w_1 - w^*\|^2 \right].$$

Since $(1 - \eta(1-\gamma)\omega) \le \exp\left(-\eta(1-\gamma)\omega\right)$,
$$\|w_T - w^*\|^2 \le \exp\left(-\eta(1-\gamma)\omega T\right)\|w_1 - w^*\|^2.$$

Then for $\epsilon$ accuracy, the gradient computation is $O(\frac{1}{\omega} \log\left(\frac{1}{\epsilon}\right))$.

$\square$

## C. Proof for Constant step-size with i.i.d. sampling

In the following sections, we assume that we have access to i.i.d. observations from the stationary distribution. We follow the vanilla TD(0) update proof in Bhandari et al. (2018).

**Theorem C.1.** *If the samples are i.i.d., let the constant step-size satisfy $\eta \leq (1 - \gamma)/8$, and we have the following convergence rate:*

$$\mathbb{E}\left[\|w_T - w^*\|^2\right] \leq \exp\left(-\eta(1-\gamma)\omega T\right)\|w_1 - w^*\|^2 + \eta\frac{2\sigma^2}{(1-\gamma)\omega}.$$

*Then to obtain accuracy $\mathbb{E}\|w_T - w^*\| \leq \epsilon$, we need total gradient computation $O\left(\left(\frac{1}{\epsilon\omega^2}\right)\log\left(\frac{1}{\epsilon}\right)\right)$ with step-size satisfying $\eta \leq \min\left\{\frac{\omega(1-\gamma)}{8}, \frac{\epsilon(1-\gamma)\omega}{2\sigma^2}\right\}.$*

*Proof.* With the update $w_{t+1} = w_t + \eta g_t(w_t)$

$$\|w_{t+1} - w^*\|^2 = \|w_t - w^*\|^2 + 2\eta g_t(w_t)^\top(w_t - w^*) + \eta^2\|g_t(w_t)\|^2.$$

Taking expectation over the i.i.d. samples, we have

$$\begin{aligned}
\mathbb{E}\left[\|w_{t+1} - w^*\|^2\right] &= \|w_t - w^*\|^2 + 2\eta\mathbb{E}_{s_t \sim \mu_\pi}\left[g_t(w_t)^\top(w_t - w^*)\right] + \eta^2\mathbb{E}\left[\|g_t(w_t)\|^2\right] \\
&\leq \|w_t - w^*\|^2 - \left(2\eta(1-\gamma) - 8\eta^2\right)\|V_{w^*} - V_{w_t}\|_D^2 + 2\eta^2\sigma^2 \quad \text{(by Lemma 3.1 Lemma 3.3)} \\
&\leq \|w_t - w^*\|^2 - \eta(1-\gamma)\|V_{w^*} - V_{w_t}\|_D^2 + 2\eta^2\sigma^2 \quad (\eta \leq (1-\gamma)/8) \\
&\leq \|w_t - w^*\|^2 - \eta(1-\gamma)\omega\|w_t - w^*\|^2 + 2\eta^2\sigma^2. \quad \text{(by Lemma 3.2)}
\end{aligned}$$

Taking expectation over $t \in [T]$, we have

$$\mathbb{E}\left[\|w_T - w^*\|^2\right] \leq (1 - \eta(1-\gamma)\omega)^T\left[\|w_1 - w^*\|^2\right] + 2\eta^2\sigma^2\sum_{t=0}^\infty(1 - \eta(1-\gamma)\omega)^t$$

Since $(1 - \eta(1-\gamma)\omega) \leq \exp\left(-\eta(1-\gamma)\omega\right)$,

$$\mathbb{E}\left[\|w_T - w^*\|^2\right] \leq \exp\left(-\eta(1-\gamma)\omega T\right)\|w_1 - w^*\|^2 + \eta\frac{2\sigma^2}{(1-\gamma)\omega}.$$

Then for $\epsilon$ accuracy, select $\eta \leq \min\left\{\frac{\omega(1-\gamma)}{8}, \frac{\epsilon(1-\gamma)\omega}{2\sigma^2}\right\}$, the gradient computation is $O\left(\frac{1}{\eta(1-\gamma)\omega}\log\left(\frac{1}{\epsilon}\right)\right) = O(\frac{1}{\epsilon\omega^2}\log\left(\frac{1}{\epsilon}\right))$. $\square$

## D. Proof for Exponential step-size with i.i.d. sampling

We now complete the proof of Theorem 3.4.

**Theorem 3.4.** *Under Assumption 2.1 and 2.2, TD(0) under i.i.d. sampling from the stationary distribution with $\eta_t = \eta_0 \alpha_t$, where $\eta_0 = \frac{1-\gamma}{8}$, $\alpha_t = \alpha^t = \frac{1}{T}^{t/T}$, $\alpha = \frac{1}{T}^{1/T}$, has the following convergence:*

$$\mathbb{E}\left[\|w_{T+1} - w^*\|^2\right]$$
$$\leq \|w_1 - w^*\|^2 e \exp\left(-\eta_0 \omega (1-\gamma) \frac{\alpha T}{\ln T}\right)$$
$$+ \frac{8\sigma^2}{e\left(\omega(1-\gamma)\right)^2} \frac{\ln^2 T}{\alpha^2 T},$$

*where $\sigma^2 = \mathbb{E}\left[\|g_t(w^*)\|^2\right]$.*

*Proof.* With the update $w_{t+1} = w_t + \eta_t g_t(w_t)$

$$\|w_{t+1} - w^*\|^2 = \|w_t - w^*\|^2 + 2\eta_t g_t(w_t)^\top (w_t - w^*) + \eta_t^2 \|g_t(w_t)\|^2.$$

Taking expectation over the i.i.d. samples, we have

$$\mathbb{E}\left[\|w_{t+1} - w^*\|^2\right] = \|w_t - w^*\|^2 + 2\eta_t \mathbb{E}_{s_t \sim \mu_\pi, s_{t+1} \sim P_\pi \mu_\pi}\left[g_t(w_t)^\top (w_t - w^*)\right] + \eta_t^2 \mathbb{E}_{s_t \sim \mu_\pi, s_{t+1} \sim P_\pi \mu_\pi}\left[\|g_t(w_t)\|^2\right]$$
$$\leq \|w_t - w^*\|^2 - \left(2\eta_t(1-\gamma) - 8\eta_t^2\right)\|V_{w^*} - V_{w_t}\|_D^2 + 2\eta_t^2 \sigma^2 \qquad \text{(by Lemma 3.1 Lemma 3.3)}$$
$$\leq \|w_t - w^*\|^2 - \eta_t(1-\gamma)\|V_{w^*} - V_{w_t}\|_D^2 + 2\eta_t^2 \sigma^2 \qquad \qquad (\eta_0 \leq (1-\gamma)/8)$$
$$\leq \|w_t - w^*\|^2 - \eta_t(1-\gamma)\omega\|w_t - w^*\|^2 + 2\eta_t^2 \sigma^2. \qquad \qquad \text{(by Lemma 3.2)}$$

Taking expectation over $t \in [T]$ and unrolling, we have

$$\mathbb{E}\left[\|w_T - w^*\|^2\right] \leq \|w_0 - w^*\|^2 \prod_{t=1}^T \left(1 - \eta_0 \omega (1-\gamma)\alpha^t\right) + 2\sigma^2 \eta_0^2 \sum_{t=1}^T \alpha^{2t} \prod_{i=t+1}^T \left(1 - \eta_0 \omega (1-\gamma)\alpha^t\right)$$

$$\leq \|w_0 - w^*\|^2 \exp\left(\underbrace{-\eta_0 \omega(1-\gamma)\sum_{t=1}^T \alpha^t}_{X}\right) + 2\sigma^2 \eta_0^2 \underbrace{\sum_{t=1}^T \alpha^{2t} \exp\left(-\eta_0 \omega(1-\gamma)\sum_{i=t+1}^T \alpha^i\right)}_{Y}.$$

Applying Lemma F.1 to the first term, we have

$$\|w_0 - w^*\|^2 \exp\left(-\eta_0 \omega(1-\gamma)\sum_{t=1}^T \alpha^t\right)$$
$$\leq \|w_0 - w^*\|^2 \exp\left(-\eta_0 \omega(1-\gamma)\left(\frac{\alpha T}{\ln(T)} - \frac{1}{\ln(T)}\right)\right)$$
$$\leq \|w_0 - w^*\|^2 \exp\left(\eta_0 \omega(1-\gamma)\frac{1}{\ln(T)}\right)\exp\left(-\eta_0 \omega(1-\gamma)\frac{\alpha T}{\ln(T)}\right)$$
$$\leq \|w_0 - w^*\|^2 e \exp\left(-\eta_0 \omega(1-\gamma)\frac{\alpha T}{\ln(T)}\right)$$
$$(\eta_0 \leq 1, \omega \leq 1, (1-\gamma) \leq 1, \frac{1}{\ln(T)} \leq 1 \text{ when } T \geq 3, \text{ thus } \exp\left(\eta_0 \omega(1-\gamma)\frac{1}{\ln(T)}\right) \leq e)$$

And applying Lemma F.2 to the second term, we have

$$\sum_{t=1}^{T} \alpha^{2t} \exp\left(-\eta_0 \omega (1-\gamma) \sum_{i=t+1}^{T} \alpha^i\right)$$

$$\leq \frac{4 \exp\left(\eta_0 \omega (1-\gamma) \frac{1}{\ln(T)}\right)}{e^2 \left(\eta_0 \omega (1-\gamma)\right)^2} \frac{\ln^2(T)}{\alpha^2 T}$$

$$\leq \frac{4e}{e^2 \left(\eta_0 \omega (1-\gamma)\right)^2} \frac{\ln^2(T)}{\alpha^2 T} \qquad \left(\exp\left(\eta_0 \omega (1-\gamma) \frac{1}{\ln(T)}\right) \leq e\right)$$

$$= \frac{4}{e \left(\eta_0 \omega (1-\gamma)\right)^2} \frac{\ln^2(T)}{\alpha^2 T}$$

Putting two terms together, we have the convergence result:

$$\mathbb{E}\left[\|w_T - w^*\|^2\right] \leq \|w_0 - w^*\|^2 e \exp\left(-\eta_0 \omega (1-\gamma) \frac{\alpha T}{\ln(T)}\right) + \frac{8\sigma^2}{e \left(\omega(1-\gamma)\right)^2} \frac{\ln^2(T)}{\alpha^2 T}.$$

$\square$

## E. Proof for Exponential step-size with Markovian sampling

Here we analyze standard TD(0) and its regularized variant with an exponential step-size under Markovian sampling. We include standard TD(0) for reference and focus on regularized TD(0) because it requires no problem-dependent parameters. The regularized update at iteration $t$ is

$$g_t^r(w) = \phi(s_t)\left(r(s_t) + (\gamma\phi(s_{t+1}) - \phi(s_t))^\top w\right) - \lambda w$$
$$= g_t(w) - \lambda w.$$

The corresponding mean-path regularized update is $g^r(w) = \mathbb{E}_{s_t \sim \mu_\pi, s_{t+1} \sim P_\pi \mu_\pi}\left[\phi(s_t)\left(r(s_t) + (\gamma\phi(s_{t+1}) - \phi(s_t))^\top w\right)\right] - \lambda w = g(w) - \lambda w$, where $P_\pi$ is the transition matrix induced by $\pi$, and $\mu_\pi$ is the stationary distribution. We define $w_r^*$ as the fixed point of regularized TD(0) update satisfying $g^r(w_r^*) = 0$.

It is clear that the standard TD(0) update involving $g_t(w)$ is a special case of $g_t^r(w)$ with $\lambda = 0$. We first state the properties shared by both methods that will be used in the proofs, and then highlight the points where their analyses differ.

**Lemma E.1.** *[Lemma 3 from Bhandari et al. (2018) with regularized update] For regularized TD(0) with mean-path sampling, the following inequality holds:*

$$\langle g^r(w), w_r^* - w \rangle \geq \left[(1-\gamma)\omega + \lambda\right] \|w - w_r^*\|^2.$$

*For standard TD(0) corresponding to $\lambda = 0$, $\langle g^r(w), w_r^* - w \rangle \geq (1-\gamma)\omega \|w - w_r^*\|^2$.*

*Proof.* Define $\xi_r = (w_r^* - w)^\top \phi(s)$ and $\xi_r' = (w_r^* - w)^\top \phi(s_{t+1})$. Since both $s$ and $s'$ are sampled from the stationary distribution, $\xi_r$ and $\xi_r'$ have the same marginal distribution. Using the expression for $g^r$,

$$g^r(w) = g^r(w) - g^r(w_r^*) \qquad (\text{since } g^r(w_r^*) = 0)$$
$$= g(w) - g(w_r^*) - \lambda(w - w_r^*) \qquad (\text{by definition of } g^r)$$
$$= \mathbb{E}_{s_t \sim \mu_\pi}\left[\phi(s_t)(\gamma\phi(s_{t+1}) - \phi(s_t))(w - w_r^*)\right] - \lambda(w - w_r^*) \qquad (\text{by definition of } g)$$
$$= \mathbb{E}_{s_t \sim \mu_\pi}\left[\phi(s)(\xi_r - \gamma\xi_r')\right] - \lambda(w - w_r^*).$$

Therefore

$$
\begin{aligned}
\langle g^r(w), w_r^* - w \rangle &= \mathbb{E}_{s_t \sim \mu_\pi, s_{t+1} \sim P_\pi \mu_\pi} \left[ \xi_r (\xi_r - \gamma \xi_r') \right] + \lambda \left\| w - w_r^* \right\|^2 && \text{(since } \xi_r = \langle \phi(s), w_r^* - w \rangle) \\
&= \mathbb{E}_{s_t \sim \mu_\pi} \left[ \xi_r^2 \right] - \gamma \mathbb{E}_{s_t \sim \mu_\pi, s_{t+1} \sim P_\pi \mu_\pi} \left[ \xi_r \xi_r' \right] + \lambda \left\| w - w_r^* \right\|^2 \\
&\geq \mathbb{E}_{s_t \sim \mu_\pi} \left[ \xi_r^2 \right] - \gamma \sqrt{\mathbb{E}_{s_t \sim \mu_\pi} \left[ \xi_r^2 \right]} \sqrt{\mathbb{E}_{s_{t+1} \sim \mu_\pi} \left[ (\xi_r')^2 \right]} + \lambda \left\| w - w_r^* \right\|^2 && \text{(by Cauchy-Schwarz)} \\
&= \mathbb{E}_{s_t \sim \mu_\pi} \left[ \xi_r^2 \right] - \gamma \mathbb{E}_{s_t \sim \mu_\pi} \left[ \xi_r^2 \right] + \lambda \left\| w - w_r^* \right\|^2 && \text{(since } \xi_r \text{ and } \xi_r' \text{ have the same marginal distribution)} \\
&= (1 - \gamma)(w_r^* - w)^\top \mathbb{E}_{s_t \sim \mu_\pi} \left[ \phi(s_t) \phi(s_t)^\top \right] (w_r^* - w) + \lambda \left\| w - w_r^* \right\|^2 && \text{(by the definition of } \xi_r) \\
&\geq ((1 - \gamma)\omega + \lambda) \left\| w - w_r^* \right\|^2 . && \text{(since } \mathbb{E}_{s_t \sim \mu_\pi} \left[ \phi(s_t) \phi(s_t)^\top \right] \succeq \omega I_d)
\end{aligned}
$$

$\square$

**Lemma E.2.** *[Lemma 4 from Bhandari et al. (2018) with regularized update] For regularized TD(0) with mean-path sampling,*

$$
\left\| g^r(w) \right\|^2 \leq (8 + 2\lambda^2) \left\| w - w_r^* \right\|^2 .
$$

*For standard TD(0) corresponding to $\lambda = 0$, $\left\| g_r(w) \right\|^2 \leq 8 \left\| w - w_r^* \right\|^2$.*

*Proof.* Similar to the proof of Lemma E.1, define $\xi_r = (w_r^* - w)^\top \phi(s_t)$ and $\xi_r' = (w_r^* - w)^\top \phi(s_{t+1})$, and note that $\xi_r$ and $\xi_r'$ have the same marginal distribution.

$$
\begin{aligned}
&\left\| g^r(w) \right\|^2 \\
&= \left\| g^r(w) - g^r(w_r^*) \right\|^2 && \text{(since } g^r(w_r^*) = 0) \\
&= \left\| g(w) - g(w_r^*) - \lambda(w - w_r^*) \right\|^2 && \text{(by the definition of } g^r) \\
&\leq 2 \left\| g(w) - g(w_r^*) \right\|^2 + 2\lambda^2 \left\| w - w_r^* \right\|^2 && \text{(since } (a + b)^2 \leq 2a^2 + 2b^2) \\
&= 2 \left\| \mathbb{E}_{s_t \sim \mu_\pi} \left[ \phi(s_t)(\xi_r - \gamma \xi_r') \right] \right\|^2 + 2\lambda^2 \left\| w - w_r^* \right\|^2 && \text{(by the definition of } g) \\
&\leq 2 \left( \sqrt{\mathbb{E}_{s_t \sim \mu_\pi} \left[ \left\| \phi(s_t) \right\|^2 \right]} \sqrt{\mathbb{E}_{s_t \sim \mu_\pi} \left[ (\xi_r - \gamma \xi_r')^2 \right]} \right)^2 + 2\lambda^2 \left\| w - w_r^* \right\|^2 && \text{(by Cauchy-Schwarz)} \\
&= 2 \mathbb{E}_{s_t \sim \mu_\pi} \left[ \left\| \phi(s_t) \right\|^2 \right] \mathbb{E}_{s_t \sim \mu_\pi} \left[ (\xi_r - \gamma \xi_r')^2 \right] + 2\lambda^2 \left\| w - w_r^* \right\|^2 \\
&\leq 2 \left( 2 \mathbb{E}_{s_t \sim \mu_\pi} \left[ \xi_r^2 \right] + 2\gamma^2 \mathbb{E}_{s_t \sim \mu_\pi} \left[ (\xi_r')^2 \right] \right) + 2\lambda^2 \left\| w - w_r^* \right\|^2 && \text{(since } \left\| \phi(s_t) \right\|^2 \leq 1 \text{ and } (a + b)^2 \leq 2a^2 + 2b^2) \\
&= 2 \left( 2(1 + \gamma^2)(w_r^* - w)^\top \mathbb{E}_{s_t \sim \mu_\pi} \left[ \phi(s_t) \phi(s_t)^\top \right] (w_r^* - w) \right) + 2\lambda^2 \left\| w - w_r^* \right\|^2 \\
& && \text{(since } \xi_r \text{ and } \xi_r' \text{ have the same marginal distribution)} \\
&\leq (8 + 2\lambda^2) \left\| w - w_r^* \right\|^2 . && \text{(since } \left\| \phi \right\|^2 \leq 1 \text{ and } \gamma \leq 1)
\end{aligned}
$$

$\square$

**Lemma E.3.** *The distance between the fixed points of regularized and standard TD(0) is bounded by:*

$$
\left\| w^* - w_r^* \right\| \leq \frac{\lambda \left\| w^* \right\|}{\lambda + \omega(1 - \gamma)} .
$$

*For standard TD(0) corresponding to $\lambda = 0$, $\left\| w^* - w_r^* \right\| = 0$.*

*Proof.* By Lemma 3.1,

$$
\begin{aligned}
(w^* - w)^\top g(w) &\geq (1 - \gamma) \left\| V_w - V_{w^*} \right\|_D^2 \\
&\geq (1 - \gamma)\omega \left\| w - w^* \right\|^2 && \text{(since } \left\| V_w - V_{w^*} \right\|_D^2 = \left\| \Phi^\top (w - w^*) \right\|_D^2 \geq \omega \left\| w - w^* \right\|^2)
\end{aligned}
$$

Substituting $w = w_r^*$, we obtain

$$(w^* - w_r^*)^\top g(w_r^*) \geq (1 - \gamma)\omega \|w_r^* - w^*\|^2.$$

By the definition of $w_r^*$, we have $g_r(w_r^*) = g(w_r^*) - \lambda w_r^* = 0$, thus $g(w_r^*) = \lambda w_r^*$. Using this relation with the above inequality,

$$
\begin{aligned}
& \lambda(w^* - w_r^*)^\top w_r^* \geq (1 - \gamma)\omega \|w_r^* - w^*\|^2 \\
\implies\ & \lambda(w^* - w_r^*)^\top (w_r^* - w^*) + \lambda(w^* - w_r^*)^\top w^* \geq (1 - \gamma)\omega \|w_r^* - w^*\|^2 && \text{(adding/subtracting)} \\
\implies\ & -\lambda \|w^* - w_r^*\|^2 + \lambda(w^* - w_r^*)^\top w^* \geq (1 - \gamma)\omega \|w_r^* - w^*\|^2 \\
\implies\ & [\lambda + \omega(1 - \gamma)] \|w^* - w_r^*\|^2 \leq \lambda(w^* - w_r^*)^\top w^* \leq \lambda \|w^* - w_r^*\| \|w^*\| && \text{(by Cauchy-Schwarz)} \\
\implies\ & \|w^* - w_r^*\| \leq \frac{\lambda \|w^*\|}{\lambda + \omega(1 - \gamma)}.
\end{aligned}
$$

$\square$

The following two lemmas are analogous to Equations 7 and 8 in Mitra (2025).

**Lemma 4.2.** *For stochastic update $g_t^\tau$, we have*

$$\|g_t^r(w)\| \leq (2 + \lambda) \|w - w_r^*\| + (3 + \lambda)\zeta,$$

*where $\zeta = \max\{1, \|w_r^*\|\}$.*

*For standard TD(0) corresponding to $\lambda = 0$, $\|g_t^r(w)\| \leq 2 \|w - w_r^*\| + 3\zeta$, $\zeta = \max\{1, \|w^*\|\}$.*

*Proof.*

$$
\begin{aligned}
\|g_t^r(w)\| &= \|g_t^r(w) - g_t^r(w_r^*) + g_t^r(w_r^*)\| && \text{(add/subtract)} \\
&= \|g_t(w) - g_t(w_r^*) - \lambda(w - w_r^*) + g_t^r(w_r^*)\| \\
&\leq \|g_t(w) - g_t(w_r^*) - \lambda(w - w_r^*)\| + \|g_t^r(w_r^*)\| && \text{(triangle inequality)} \\
&= \left\|\left(\phi(s_t)(\gamma\phi(s_{t+1}) - \phi(s_t))^\top - \lambda\right)(w - w_r^*)\right\| + \left\|\phi(s_t)\left(r(s_t) + (\gamma\phi(s_{t+1}) - \phi(s_t))^\top w_r^*\right) - \lambda w_r^*\right\| \\
&\leq \left\|\left(\phi(s_t)(\gamma\phi(s_{t+1}) - \phi(s_t))^\top\right)(w - w_r^*)\right\| + \lambda \|w - w_r^*\| \\
&\quad + \|\phi(s_t)r(s_t)\| + \left\|\phi(s_t)\left((\gamma\phi(s_{t+1}) - \phi(s_t))^\top w_r^*\right)\right\| + \lambda \|w_r^*\| && (\|a + b\| \leq \|a\| + \|b\|) \\
&\leq \left\|\left(\phi(s_t)(\gamma\phi(s_{t+1}) - \phi(s_t))^\top\right)\right\| \|w - w_r^*\| + \lambda \|w - w_r^*\| \\
&\quad + \|\phi(s_t)r(s_t)\| + \|\phi(s_t)(\gamma\phi(s_{t+1}) - \phi(s_t))\| \|w_r^*\| + \lambda \|w_r^*\| && \text{(by Cauchy-Schwarz)} \\
&\leq (2 + \lambda) \|w - w_r^*\| + (3 + \lambda)\zeta, && (r(s_t) \leq 1, \|\phi(s_t)\| \leq 1, \text{Equation (7)})
\end{aligned}
$$

where $\zeta = \max\{1, \|w_r^*\|\}$. $\square$

The corresponding bound on the mean-path update is

**Lemma 4.3.** *For mean-path update $g^\tau$, we have*

$$\|g^r(w)\| \leq (2 + \lambda) \|w - w_r^*\|,$$

*where $\zeta = \max\{1, \|w_r^*\|\}$. For standard TD(0) corresponding to $\lambda = 0$, $\|g^r(w)\| \leq 2 \|w - w_r^*\|$.*

*Proof.*

$$
\begin{aligned}
\|g^r(w)\| &= \|g^r(w) - g^r(w_r^*)\| && (g^r(w_r^*) = 0) \\
&= \|(g(w) - g(w_r^*)) - \lambda(w - w_r^*)\| && \text{(by definition of } g^r(w)) \\
&\leq \|g(w) - g(w_r^*)\| + \lambda \|w - w_r^*\| && \text{(triangle inequality)} \\
&\leq (2 + \lambda) \|w - w_r^*\|. && \text{(by Equation (6))}
\end{aligned}
$$

$\square$

In the Markovian case, we also need the definition of mixing time. We restate it below and provide a proof.

**Definition 4.1.** Define the mixing time as $\tau_\delta = \min\{t \in \mathbb{N}_0 \mid m\rho^t \leq \delta\}$, where $\delta \in (0,1)$.

**Lemma 4.4.** *For any initial state distribution $\mu_0$, the state distribution at time $t$ is $P_\pi^t \mu_0$. For any $w$, when $t \geq \tau_\delta$,*

$$\left\| \mathbb{E}_{s_t \sim P_\pi^t \mu_0} [g_t^r(w)] - g^r(w) \right\| \leq 2(2+\lambda)\delta \left( \|w\| + 1 \right).$$

*For standard TD(0) corresponding to setting $\lambda = 0$,*

$$\left\| \mathbb{E}_{s_t \sim P_\pi^t \mu_0} [g_t(w)] - g(w) \right\| \leq 4\delta \|w\| + 1.$$

*Proof.* Let $\|g_t^r(w)\|_\infty$ denote the supremum of $\|g_t^r(w)\|$ over states.

$$
\begin{aligned}
\|g_t^r(w)\|_\infty &= \max_{s_t \in \mathcal{S}} \|g_t^r(w)\| \\
&= \max_{s_t \in \mathcal{S}} \left\| \left( r(s_t) + \gamma w^\top \phi(s_t') - w^\top \phi(s_t) \right) \phi(s_t) - \lambda w \right\| && \text{(Definition of } g_t^r(w)) \\
&\leq (2+\lambda)\|w\| + 1. && \text{(since } r(s_t) \leq 1, \|\phi(s_t)\| \leq 1, \gamma \leq 1)
\end{aligned}
$$

With $\tau_\delta = \min\{t \in \mathbb{N}_0 \mid m\rho^t \leq \delta\}$, and $P_\pi^t \mu_0$ representing the probability distribution over the states after $t$ steps with initial state distribution $\mu_0$, we have

$$
\begin{aligned}
\left\| \mathbb{E}_{s_t \sim P_\pi^t \mu_0} [g_t^r(w)] - g^r(w) \right\| &= \left\| \mathbb{E}_{s_t \sim P_\pi^t \mu_0} [g_t^r(w)] - \mathbb{E}_{s_t \sim \mu_\pi} [g_t^r(w)] \right\| && \text{(Definition of } g_t^r(w) \text{ and } g^r(w)) \\
&= \left\| \sum_{s_t \in \mathcal{S}} g_t^r(w) \left( (P_\pi^t \mu_0)(s_t) - \mu_\pi(s_t) \right) \right\| \\
&\leq \sum_{s_t \in \mathcal{S}} \left\| g_t^r(w) \left( (P_\pi^t \mu_0)(s_t) - \mu_\pi(s_t) \right) \right\| && \text{(triangle inequality)} \\
&\leq \|g_t^r(w)\|_\infty \sum_{s_t \in \mathcal{S}} \left| (P_\pi^t \mu_0)(s_t) - \mu_\pi(s_t) \right| && (\|g_t^r(w)\| \leq \|g_t^r(w)\|_\infty) \\
&\leq 2 \|g_t^r(w)\|_\infty \sup_{\mu_0} d_{\text{TV}} \left( P_\pi^t \mu_0, \mu_\pi \right) \\
& \text{(by the definition of the total variation distance, and taking the sup over } s_0) \\
&\leq 2 \|g_t^r(w)\|_\infty m\rho^t && \text{(using Equation (2))} \\
&\leq 2m\rho^t ((2+\lambda)\|w\| + 1) && \text{(using the bound on } \|g_t^r(w)\|_\infty) \\
&\leq 2(2+\lambda)\delta(\|w\| + 1) && \text{(using the definition of } t_\delta)
\end{aligned}
$$

$\square$

**Lemma E.4.** *For $\tau_{mix}$ defined in Equation (3),*

$$\tau_{mix} = a \ln(T') + b,$$

*where $a$ and $b$ are constants with $a = \frac{1}{\ln(1/\rho)}$, $b = \frac{\ln(2(2+\lambda)m)}{\ln(1/\rho)}$ and $T' = \frac{T}{\eta_0}$.*

*Proof.* By Equation (2), Lemma 4.4 and the definition of $\tau_{mix}$ in Equation (3), we obtain the expression for $\tau_{mix}$:

$$
\begin{aligned}
\tau_{mix} &= \frac{\ln(2(2+\lambda)Tm/\eta_0)}{\ln(1/\rho)} \\
&= \frac{\ln(T')}{\ln(1/\rho)} + \frac{\ln(2(2+\lambda)m)}{\ln(1/\rho)} && \text{(defining } T' = \frac{T}{\eta_0})
\end{aligned}
$$

$\square$

With this expression for $\tau_{mix}$, we prove the following lemmas.

**Lemma E.5.** *If $\alpha := \frac{1}{T}^{1/T}$, $T \geq 3$, $\eta_0 \leq 1$ for $\tau_{mix}$ defined in Equation (3), then for all $t \leq \tau_{mix}$*

$$\frac{1 - \alpha^t}{1 - \alpha} \leq 4 \max\{a, b\} \ln(T'),$$

*where $a = \frac{1}{\ln(1/\rho)}$, $b = \frac{\ln(2(2+\lambda)m)}{\ln(1/\rho)}$ and $T' = \frac{T}{\eta_0}$.*

*Proof.* Using the expression for $\tau_{\text{mix}} = a \ln(T') + b$ from Lemma E.4,

$$\frac{1 - \alpha^t}{1 - \alpha} \leq \frac{1 - \alpha^{\tau_{\text{mix}}}}{1 - \alpha} = \frac{1 - (1/T)^{\tau_{\text{mix}}/T}}{1 - (1/T)^{1/T}} = \frac{1 - (1/T)^{(a \ln(T') + b)/T}}{1 - (1/T)^{1/T}} \qquad \text{(since } t \leq \tau_{\text{mix}})$$

$$= \frac{1 - \exp\left(\frac{-a(\ln(T') \ln(T))}{T} - \frac{b \ln(T)}{T}\right)}{1 - \exp\left(-\frac{\ln(T)}{T}\right)}.$$

Define $j := \frac{a \ln(T) \ln(T') + b \ln(T)}{T}$, and $k := \frac{\ln(T)}{T}$. Note that $j$ and $k$ are related by $j = a \ln(T')k + bk$. We can simplify the above inequality as follows:

$$\frac{1 - \alpha^{\tau_{\text{mix}}}}{1 - \alpha} = \frac{1 - \exp(-j)}{1 - \exp(-k)}$$

To bound the numerator and denominator separately, we use the fact that $\frac{v}{1+v} \leq 1 - \exp(-v) \leq v$ for $v > 0$. For the numerator, setting $v = j$, we have $1 - \exp(-j) \leq j$. And for the denominator, setting $v = k$, we have $1 - \exp(-k) \geq k/1+k$. Combining the above relations,

$$\implies \frac{1 - \alpha^t}{1 - \alpha} \leq \frac{j(1 + k)}{k}$$

$$= \frac{(a \ln(T')k + bk)(1 + k)}{k} \qquad \text{(since } j = a \ln(T')k + bk)$$

$$= (a \ln(T') + b)\left(1 + \frac{\ln(T)}{T}\right) \qquad \text{(since } k = \ln(T)/T)$$

$$\leq (a \ln(T') + b)(1 + 1/e) \qquad (\tfrac{\ln(T)}{T} \text{ decreases after } T = e, \text{ assuming } T \geq 3)$$

$$\leq 4 \max\{a, b\} \ln(T'). \qquad \text{(since } \eta_0 \leq 1 \text{ and } T \geq 1 \implies T' \geq 1)$$

$\square$

**Lemma E.6.** *If $\alpha := \frac{1}{T}^{1/T}$, $T \geq \max\left\{3, \frac{1}{\eta_0}\right\}$, and $T$ is large enough that $\frac{\ln^2(T)}{T} \leq \frac{1}{2a}$ and $\frac{\ln(T)}{T} \leq \frac{1}{b}$, with $\eta_0 \leq 1$, for $\tau_{mix}$ defined in Equation (3) and for all $t \leq \tau_{mix}$,*

$$\frac{\alpha^{-\tau_{mix}} - 1}{1 - \alpha} \leq 8 \max\{a, b\} \ln(T')$$

*where $a = \frac{1}{\ln(1/\rho)}$, $b = \frac{\ln(2(2+\lambda)m)}{\ln(1/\rho)}$ and $T' = \frac{T}{\eta_0}$.*

*Proof.* First note that,

$$\alpha = \left(\frac{1}{T}\right)^{1/T} = \exp\left(-\frac{1}{T} \ln(T)\right)$$

Using the expression for $\tau_{\text{mix}} = a \ln(T') + b$ from Lemma E.4,

$$\alpha^{\tau_{\text{mix}}} = \exp\left(-\frac{\tau_{\text{mix}}}{T} \ln(T)\right) = \exp\left(-\frac{a \ln(T') + b}{T} \ln(T)\right)$$

Define $k := \frac{\ln(T)}{T}$ and $j := [a \ln(T') + b] \frac{\ln(T)}{T} = [a \ln(T') + b] k$. Using the above relations,

$$\frac{\alpha^{-\tau_{\mathrm{mix}}} - 1}{1 - \alpha} = \frac{\exp(j) - 1}{1 - \exp(-k)}$$

In order to simplify the above expression, we use the following inequalities:

$$\forall x \geq 0 \,, 1 - \exp(-x) \geq \frac{x}{1 + x} \quad ; \quad \forall y \in (0, 1) \,, \exp(2y) \leq \frac{1 + y}{1 - y}$$

Since $T \geq 1$, $k \geq 0$ and hence we can use $x = k$ to conclude that $1 - \exp(-k) \geq \frac{k}{1+k}$.

We substitute $y$ with $j/2$, which requires $0 < j < 2$. $j = [a \ln(T') + b] \frac{\ln(T)}{T} = \tau_{\mathrm{mix}} \frac{\ln(T)}{T} > 0$ is already satisfied. Ensuring $j \leq 2$ requires that $[a \ln(T') + b] \frac{\ln(T)}{T} \leq 2$. For $\eta_0 \leq 1$ and $T \geq 1$, $T' \geq 1$. Moreover, for $T \geq \frac{1}{\eta_0}$, $\frac{\ln(T')}{\ln(T)} \leq 2$. Hence, it suffices to ensure that

$$2 \, a \, \frac{\ln^2(T)}{T} + b \, \frac{\ln(T)}{T} \leq 2$$

Therefore, it suffices to ensure that

$$\frac{\ln^2(T)}{T} \leq \frac{1}{2a} \quad \text{and} \quad \frac{\ln(T)}{T} \leq \frac{1}{b}$$

With these constraints on $T$, we can guarantee that $j \leq 1$. Using $y = \frac{j}{2}$ in the above inequality, we can conclude that,

$$\exp(j) - 1 \leq \frac{1 + j/2}{1 - j/2} - 1 = \frac{j}{1 - j/2} \leq 2 \, j$$

Combining the above relations, we get that,

$$\frac{\alpha^{-\tau_{\mathrm{mix}}} - 1}{1 - \alpha} \leq \frac{2 \, j \, (k + 1)}{k}$$

Following the same steps as in the proof of Lemma E.5, we conclude that, for $T \geq 3$ and $\eta_0 \leq 1$,

$$\frac{\alpha^{-\tau_{\mathrm{mix}}} - 1}{1 - \alpha} \leq 8 \, \max\{a, b\} \, \ln(T')$$

$\square$

**Lemma 4.5.** *For the regularized TD(0) update with exponential step-sizes* $\eta_t = \eta_0 \alpha_t$, *where* $\eta_0 \leq \frac{1 - \gamma}{16 \ln(T)}$, $\alpha_t = \alpha^t = \frac{1}{T}^{t/T}$, $\alpha = \frac{1}{T}^{1/T}$, *if* $T \geq \max\{3, 1/\eta_0\}$,

$$\forall t \leq \tau_{mix}, \quad \|w_t - w_r^*\|^2 \leq B(\tau_{mix}) \quad \textbf{(Base case)},$$

*where* $B(\tau_{mix}) := \exp(2(2 + \lambda) \max\{a, b\}) \cdot \|w_1 - w_r^* + \zeta\|^2$, *where* $a = \frac{1}{\ln(1/\rho)}$, $b = \frac{\ln(2(2 + \lambda)m)}{\ln(1/\rho)}$, $\zeta = \max\{1, \|w_r^*\|\}$.

*Proof.*

$$\begin{aligned}
\|w_{t+1} - w_r^*\| &\leq \|w_t - w_r^*\| + \eta_t \|g_t^r(w_t)\| \\
&\leq (1 + (2 + \lambda)\eta_t) \|w_t - w_r^*\| + (3 + \lambda)\eta_t \zeta. \qquad \text{(by Lemma 4.2)}
\end{aligned}$$

Iterating the above inequality, we have $\forall t \leq \tau_{\mathrm{mix}}$:

$$\|w_t - w_r^*\| \leq \|w_1 - w_r^*\| \prod_{i=1}^{t}(1 + (2+\lambda)\eta_i) + (3+\lambda)\zeta \sum_{i=1}^{t} \eta_i \prod_{j=i+1}^{t}(1 + (2+\lambda)\eta_j)$$

$$\leq \|w_1 - w_r^*\| \exp\left((2+\lambda)\sum_{i=1}^{t}\eta_i\right) + (3+\lambda)\zeta \sum_{i=1}^{t}\eta_i \exp\left((2+\lambda)\sum_{j=i+1}^{t}\eta_j\right) \quad \text{(since } 1+x \leq \exp(x)\text{)}$$

$$= \|w_1 - w_r^*\| \exp\left((2+\lambda)\eta_0 \sum_{i=1}^{t}\alpha^i\right) + (3+\lambda)\zeta\eta_0 \sum_{i=1}^{t}\alpha^i \exp\left((2+\lambda)\eta_0 \sum_{j=i+1}^{t}\alpha^j\right)$$

$$= \|w_1 - w_r^*\| \exp\left((2+\lambda)\eta_0 \sum_{i=1}^{t}\alpha^i\right) + (3+\lambda)\zeta\eta_0 \sum_{i=1}^{t}\alpha^i \exp\left((2+\lambda)\eta_0 \frac{\alpha^{i+1} - \alpha^{t+1}}{1-\alpha}\right)$$

$$= \|w_1 - w_r^*\| \exp\left((2+\lambda)\eta_0 \frac{\alpha - \alpha^{t+1}}{1-\alpha}\right) + (3+\lambda)\zeta\eta_0 \sum_{i=1}^{t}\alpha^i \exp\left((2+\lambda)\eta_0 \frac{\alpha^{i+1} - \alpha^{t+1}}{1-\alpha}\right)$$

$$\leq \|w_1 - w_r^*\| \exp\left((2+\lambda)\eta_0 \frac{\alpha - \alpha^{t+1}}{1-\alpha}\right) + (3+\lambda)\zeta\eta_0 \sum_{i=1}^{t}\alpha^i \exp\left((2+\lambda)\eta_0 \frac{\alpha - \alpha^{t+1}}{1-\alpha}\right)$$

$$\text{(since } \alpha^{i+1} \leq \alpha\text{)}$$

$$= \exp\left((2+\lambda)\eta_0 \frac{\alpha - \alpha^{t+1}}{1-\alpha}\right) \left[\|w_1 - w_r^*\| + (3+\lambda)\zeta\eta_0 \sum_{i=1}^{t}\alpha^i\right]$$

$$= \exp\left((2+\lambda)\eta_0\alpha \frac{1-\alpha^t}{1-\alpha}\right) \left[\|w_1 - w_r^*\| + (3+\lambda)\zeta\eta_0\alpha \frac{1-\alpha^t}{1-\alpha}\right]$$

$$\leq \exp(4(2+\lambda)\eta_0\alpha \max\{a,b\}\ln(T')) \left[\|w_1 - w_r^*\| + 4(3+\lambda)\zeta\eta_0\alpha \max\{a,b\}\ln(T')\right] \quad \text{(by Lemma E.5)}$$

since $\eta_0 \leq \frac{1-\gamma}{16\ln(T)}$,

$$\leq \exp\left(\frac{1}{4}(2+\lambda)\alpha(1-\gamma)\max\{a,b\}\frac{\ln(T')}{\ln(T)}\right) \left[\|w_1 - w_r^*\| + \frac{1}{4}(3+\lambda)\alpha(1-\gamma)\max\{a,b\}\frac{\ln(T')}{\ln(T)}\zeta\right]$$

$$\leq \exp\left(\frac{1}{2}(2+\lambda)\alpha(1-\gamma)\max\{a,b\}\right) \left[\|w_1 - w_r^*\| + \frac{1}{2}(3+\lambda)\alpha(1-\gamma)\max\{a,b\}\zeta\right]$$

$$\text{(for } T \geq \frac{1}{\eta_0}, \frac{\ln(T')}{\ln(T)} \leq 2\text{)}$$

$$\leq \exp\left(\frac{1}{2}(2+\lambda)\max\{a,b\}\right) \exp\left(\frac{1}{4}(3+\lambda)\max\{a,b\}\right) \left[\|w_1 - w_r^*\| + \zeta\right]$$

$$\text{(} \alpha \leq 1, (1-\gamma) \leq 1, \max\{a,b\} \leq \exp\left(\tfrac{1}{2}\max\{a,b\}\right)\text{)}$$

$$\leq \exp((2+\lambda)\max\{a,b\}) \left[\|w_1 - w_r^*\| + \zeta\right].$$

Squaring the both sides, we get:

$$\|w_t - w_r^*\|^2 \leq \exp(2(2+\lambda)\max\{a,b\}) \left[\|w_1 - w_r^*\| + \zeta\right]^2.$$

Let $B(\tau_{\mathrm{mix}}) := \exp(2(2+\lambda)\max\{a,b\}) \left[\|w_1 - w_r^*\| + \zeta\right]^2$, we have that $\|w_t - w_r^*\|^2 \leq B(\tau_{\mathrm{mix}})$ for all $t \leq \tau_{\mathrm{mix}}$. $\qquad \square$

Using the above lemmas, we will follow a proof similar to that in Mitra (2025). For this, we define the following notation:

$$d_t := \mathbb{E}\left[\|w_t - w_r^*\|\right]^2,$$

and

$$e_t := \mathbb{E}\left[\langle w_t - w_r^*, g_t^r(w_t) - g^r(w_t)\rangle\right],$$

which includes the error introduced by sampling along the Markov chain. The expectations are taken with respect to state distribution at time $t$. For the subsequent lemmas, we omit the subscript $P_\pi^t \mu_0$ for brevity.

**Lemma 4.6.** *Let $T \geq \max\{3, \frac{1}{\eta_0}\}$, and let $T$ be large enough that $\frac{\ln^2(T)}{T} \leq \frac{1}{2a}$, $\frac{\ln(T)}{T} \leq \frac{1}{b}$, and $\ln(T) \geq \max\{a, b\}$. Suppose for all $t \geq \tau_{mix}$, if $\|w_k - w_r^*\|^2 \leq B(\tau_{mix})$ for all $k \in [t]$, then,*

$$\|w_t - w_{t-\tau_{mix}}\|^2 \leq c_1^2 \, B(\tau_{mix}) \, \eta_t^2 \, \ln^4(T),$$

*where $c_1^2 = 2560(2 + \lambda)^2$, $a = \frac{1}{\ln(1/\rho)}$ and $b = \frac{\ln(2(2+\lambda)m)}{\ln(1/\rho)}$.*

*Proof.*

$$
\begin{aligned}
\|w_t - w_{t-\tau_{mix}}\| &\leq \sum_{i=t-\tau_{mix}}^{t-1} \|w_{i+1} - w_i\| && \text{(triangle inequality)} \\
&\leq \sum_{i=t-\tau_{mix}}^{t-1} \eta_i \, \|g_i^r(w_i)\| && \text{(by the update)} \\
&\leq \sum_{i=t-\tau_{mix}}^{t-1} \eta_i((2+\lambda)\|w_i - w_r^*\| + (3+\lambda)\zeta) && \text{(by Lemma 4.2)} \\
&\leq \sum_{i=t-\tau_{mix}}^{t-1} \eta_i\Big((2+\lambda)\sqrt{B(\tau_{mix})} + (3+\lambda)\zeta\Big)
\end{aligned}
$$

$$\text{(assuming that } \|w_k - w_r^*\|^2 \leq B(\tau_{mix}) \text{ for } k \in [t])$$

$$
\begin{aligned}
&= \underbrace{\Big((2+\lambda)\sqrt{B(\tau_{mix})} + (3+\lambda)\zeta\Big)}_{:=C} \, \eta_0 \sum_{i=t-\tau_{mix}}^{t-1} \alpha^i && \text{(by definition of the exponential step-sizes)} \\
&= C \, \eta_0 \, \alpha^{t-\tau_{mix}} \sum_{i=0}^{\tau_{mix}-1} \alpha^i \\
&= C \, \eta_0 \, \frac{\alpha^t}{\alpha^{\tau_{mix}}} \frac{1 - \alpha^{\tau_{mix}}}{1 - \alpha} = C \, \eta_t \, \frac{\alpha^{-\tau_{mix}} - 1}{1 - \alpha} \\
&\leq 8 \, C \, \eta_t \, \max\{a, b\} \, \ln(T') && \text{(using Lemma E.6)}
\end{aligned}
$$

$$
\begin{aligned}
\implies \|w_t - w_{t-\tau_{mix}}\|^2 &\leq 64 \, C^2 \, \eta_t^2 \, [\max\{a, b\}]^2 \, \ln^2(T') \\
&= 64 \Big((2+\lambda)\sqrt{B(\tau_{mix})} + (3+\lambda)\zeta\Big)^2 \eta_t^2 \, [\max\{a, b\}]^2 \, \ln^2(T') \\
&\leq 256 \Big((2+\lambda)\sqrt{B(\tau_{mix})} + (3+\lambda)\zeta\Big)^2 \eta_t^2 \, [\max\{a, b\}]^2 \, \ln^2(T)
\end{aligned}
$$

$$\text{(for } \eta_0 \leq 1 \text{ and } T \geq 1, \frac{\ln(T')}{\ln(T)} \leq 2)$$

$$
\begin{aligned}
&\leq 256 \left[2(2+\lambda)^2 B(\tau_{mix}) + 2(3+\lambda)^2 \zeta^2\right] \eta_t^2 \, [\max\{a, b\}]^2 \, \ln^2(T) \\
&&&\hspace{-4cm}\text{(since } (x+y)^2 \leq 2x^2 + 2y^2) \\
&\leq 256 \left[2(2+\lambda)^2 B(\tau_{mix}) + 2(3+\lambda)^2 \zeta^2\right] \eta_t^2 \, \ln^4(T) && \text{(since } \ln(T) \geq \max\{a, b\}) \\
&\leq \underbrace{2560(2+\lambda)^2}_{:=c_1^2} B(\tau_{mix}) \, \eta_t^2 \, \ln^4(T) && \text{(since } \zeta^2 \leq B(\tau_{mix}) \text{ and } 2(3+\lambda)^2 \leq 8(2+\lambda)^2) \\
&= c_1^2 \, B(\tau_{mix}) \, \eta_t^2 \, \ln^4(T)
\end{aligned}
$$

$\square$

**Lemma 4.7.** *Let $T \geq \max\{3, \frac{1}{\eta_0}\}$, and let $T$ be large enough that $\frac{\ln^2(T)}{T} \leq \frac{1}{2a}$, $\frac{\ln(T)}{T} \leq \frac{1}{b}$, and $\ln(T) \geq \max\{a, b\}$. For $t \geq \tau_{mix}$, suppose $\|w_k - w_r^*\|^2 \leq B(\tau_{mix})$ for all $k \in [t]$. Then:*

$$
\mathbb{E}_t\left[\langle g_t^r(w_t) - g^r(w_t), \, w_t - w_r^* \rangle\right]
$$
$$
\leq C \, \eta_t \, \ln^2(T) \, B(\tau_{mix}),
$$

where $C = C_1 + 3 + 2C_2$, $C_1 = \frac{c_1}{2}$ and $C_2 = \frac{c_1 c_2}{2}$, $c_1 = 2560(2+\lambda)^2$ and $c_2 = 4(2+\lambda)^2 + 4(3+\lambda)^2 + 2(2+\lambda)^2$.

*Proof.* Following Mitra (2025), we decompose as: $\langle w_t - w_r^*, g_t^r(w_t) - g^r(w_t) \rangle = T_1 + T_2 + T_3 + T_4$, where

$$
\begin{aligned}
T_1 &= \langle w_t - w_{t-\tau_{\mathrm{mix}}}, g_t^r(w_t) - g^r(w_t) \rangle, \\
T_2 &= \langle w_{t-\tau_{\mathrm{mix}}} - w_r^*, g_t^r(w_{t-\tau_{\mathrm{mix}}}) - g^r(w_{t-\tau_{\mathrm{mix}}}) \rangle, \\
T_3 &= \langle w_{t-\tau_{\mathrm{mix}}} - w_r^*, g_t^r(w_t) - g_t^r(w_{t-\tau_{\mathrm{mix}}}) \rangle, \quad \text{and} \\
T_4 &= \langle w_{t-\tau_{\mathrm{mix}}} - w_r^*, g^r(w_{t-\tau_{\mathrm{mix}}}) - g^r(w_t) \rangle.
\end{aligned}
$$

For $T_1$:

$$
\begin{aligned}
T_1 &\leq \|w_t - w_{t-\tau_{\mathrm{mix}}}\| \, \|g_t^r(w_t) - g^r(w_t)\| && \text{(by Cauchy-Schwarz)} \\
&\leq \frac{1}{2 c_1 \eta_t \ln^2(T)} \|w_t - w_{t-\tau_{\mathrm{mix}}}\|^2 + \frac{c_1 \eta_t \ln^2(T)}{2} \|g_t^r(w_t) - g^r(w_t)\|^2 && \text{(by Young's inequality)} \\
&\leq \frac{c_1 \eta_t \ln^2(T) B(\tau_{\mathrm{mix}})}{2} + \frac{c_1 \eta_t \ln^2(T)}{2} \|g_t^r(w_t) - g^r(w_t)\|^2 && \text{(using Lemma 4.6)}
\end{aligned}
$$

Simplifying $\|g_t^r(w_t) - g^r(w_t)\|^2$,

$$
\begin{aligned}
\|g_t^r(w_t) - g^r(w_t)\|^2 &\leq 2\|g_t^r(w_t)\|^2 + 2\|g^r(w_t)\|^2 && \text{(since } (x+y)^2 \leq 2x^2 + 2y^2) \\
&\leq 2[(2+\lambda)\|w_t - w_r^*\| + (3+\lambda)\zeta]^2 + 2[(2+\lambda)\|w_t - w_r^*\|]^2 \\
&&& \text{(using Lemma 4.2 and Lemma 4.3)} \\
&\leq 4(2+\lambda)^2 \|w_t - w_r^*\|^2 + 4(3+\lambda)^2 \zeta^2 + 2(2+\lambda)^2 \|w_t - w_r^*\|^2 \\
&&& \text{(since } (x+y)^2 \leq 2x^2 + 2y^2) \\
&\leq 4(2+\lambda)^2 B(\tau_{\mathrm{mix}}) + 4(3+\lambda)^2 B(\tau_{\mathrm{mix}}) + 2(2+\lambda)^2 B(\tau_{\mathrm{mix}}) \\
&&& \text{(since } d_k \leq B(\tau_{\mathrm{mix}}) \text{ for all } k \in [t] \text{ and } \zeta^2 \leq B(\tau_{\mathrm{mix}})) \\
&= \underbrace{(4(2+\lambda)^2 + 4(3+\lambda)^2 + 2(2+\lambda)^2)}_{:=c_2} B(\tau_{\mathrm{mix}}) \\
\implies \|g_t^r(w_t) - g^r(w_t)\|^2 &\leq c_2 B(\tau_{\mathrm{mix}})
\end{aligned}
$$

Combining the above inequalities,

$$
T_1 \leq \frac{c_1 \eta_t \ln^2(T) B(\tau_{\mathrm{mix}})}{2} + \frac{c_1 c_2 \eta_t \ln^2(T)}{2} B(\tau_{\mathrm{mix}}) = \eta_t \ln^2(T) B(\tau_{\mathrm{mix}}) \underbrace{\left[\frac{c_1}{2} + \frac{c_1 c_2}{2}\right]}_{:=C_1}
$$

$$
\implies T_1 \leq C_1 \eta_t \ln^2(T) B(\tau_{\mathrm{mix}})
$$

For $T_3$:

$$
\begin{aligned}
T_3 &\leq \|w_{t-\tau_{\text{mix}}} - w_r^*\| \, \|g_t^r(w_t) - g_t^r(w_{t-\tau_{\text{mix}}})\| && \text{(by Cauchy-Schwarz)} \\
&= \|w_{t-\tau_{\text{mix}}} - w_r^*\| \, \|g_t(w_t) - g_t(w_{t-\tau_{\text{mix}}}) - \lambda(w_t - w_{t-\tau_{\text{mix}}})\| && \text{(by definition)} \\
&\leq \|w_{t-\tau_{\text{mix}}} - w_r^*\| \, (\|g_t(w_t) - g_t(w_{t-\tau_{\text{mix}}})\| + \lambda \|w_t - w_{t-\tau_{\text{mix}}}\|) && \text{(triangle inequality)} \\
&\leq \|w_{t-\tau_{\text{mix}}} - w_r^*\| \, (2\|w_t - w_{t-\tau_{\text{mix}}}\| + \lambda \|w_t - w_{t-\tau_{\text{mix}}}\|) && \text{(by Equation (7))} \\
&= \|w_{t-\tau_{\text{mix}}} - w_r^*\| \, (2+\lambda) \, \|w_t - w_{t-\tau_{\text{mix}}}\| \\
&\leq \frac{1}{2 c_1 \eta_t \ln^2(T)} \|w_t - w_{t-\tau_{\text{mix}}}\|^2 + \frac{c_1 \eta_t (2+\lambda) \ln^2(T)}{2} \|w_{t-\tau_{\text{mix}}} - w_r^*\|^2 && \text{(by Young's inequality)} \\
&\leq \frac{c_1 \eta_t \ln^2(T) B(\tau_{\text{mix}})}{2} + \frac{c_1 \eta_t (2+\lambda) \ln^2(T)}{2} \|w_{t-\tau_{\text{mix}}} - w_r^*\|^2 && \text{(by Lemma 4.6)} \\
&\leq \frac{c_1 \eta_t \ln^2(T) B(\tau_{\text{mix}})}{2} + \frac{c_1 \eta_t (2+\lambda) \ln^2(T) B(\tau_{\text{mix}})}{2} && \text{(since } d_k \leq B(\tau_{\text{mix}}) \text{ for all } k \in [t]) \\
&= \eta_t \ln^2(T) B(\tau_{\text{mix}}) \underbrace{\left[ \frac{c_1}{2} + \frac{c_1(2+\lambda)}{2} \right]}_{:=C_2}
\end{aligned}
$$

$$
\implies T_3 \leq C_2 \, \eta_t \, \ln^2(T) \, B(\tau_{\text{mix}})
$$

For $T_4$, the same analysis applies.

$$
\begin{aligned}
T_4 &\leq \|w_{t-\tau_{\text{mix}}} - w_r^*\| \, \|g^r(w_{t-\tau_{\text{mix}}}) - g^r(w_t)\| && \text{(by Cauchy-Schwarz)} \\
&= \|w_{t-\tau_{\text{mix}}} - w_r^*\| \, \|g(w_t) - g(w_{t-\tau_{\text{mix}}}) - \lambda(w_t - w_{t-\tau_{\text{mix}}})\| && \text{(by definition)} \\
&\leq \|w_{t-\tau_{\text{mix}}} - w_r^*\| \, (2\|w_t - w_{t-\tau_{\text{mix}}}\| + \lambda \|w_t - w_{t-\tau_{\text{mix}}}\|) && \text{(by Equation (6))} \\
&= \|w_{t-\tau_{\text{mix}}} - w_r^*\| \, (2+\lambda) \, \|w_t - w_{t-\tau_{\text{mix}}}\|
\end{aligned}
$$

Following the same analysis for $T_3$, we get that,

$$
T_4 \leq C_2 \, \eta_t \, \ln^2(T) \, B(\tau_{\text{mix}})
$$

For $T_2$, following the proof in Mitra (2025):

$$
\begin{aligned}
\mathbb{E}[T_2] &= \mathbb{E}\left[ \langle w_{t-\tau_{\text{mix}}} - w_r^*, g_t^r(w_{t-\tau_{\text{mix}}}) - g^r(w_{t-\tau_{\text{mix}}}) \rangle \right] \\
&= \mathbb{E}\left[ \mathbb{E}\left[ \langle w_{t-\tau_{\text{mix}}} - w_r^*, g_t^r(w_{t-\tau_{\text{mix}}}) - g^r(w_{t-\tau_{\text{mix}}}) \rangle | w_{t-\tau_{\text{mix}}} \right] \right] \\
&= \mathbb{E}\left[ \langle w_{t-\tau_{\text{mix}}} - w_r^*, \mathbb{E}[g_t^r(w_{t-\tau_{\text{mix}}}) - g^r(w_{t-\tau_{\text{mix}}}) | w_{t-\tau_{\text{mix}}}] \rangle \right] \\
&\leq \mathbb{E}[\|w_{t-\tau_{\text{mix}}} - w_r^*\| \, \|\mathbb{E}[g_t^r(w_{t-\tau_{\text{mix}}}) - g^r(w_{t-\tau_{\text{mix}}}) \mid w_{t-\tau_{\text{mix}}}]\|] && \text{(by Cauchy-Schwarz)} \\
&\leq \eta_T \mathbb{E}[\|w_{t-\tau_{\text{mix}}} - w_r^*\|(1 + \|w_{t-\tau_{\text{mix}}}\|)] && \text{(by Equation (4))} \\
&\leq \eta_t \mathbb{E}[\|w_{t-\tau_{\text{mix}}} - w_r^*\|(1 + \|w_{t-\tau_{\text{mix}}}\|)] && \text{(exponential step-size decreases)} \\
&\leq \eta_t \mathbb{E}[\|w_{t-\tau_{\text{mix}}} - w_r^*\|(1 + \|w_r^*\| + \|w_{t-\tau_{\text{mix}}} - w_r^*\|)] && \text{(triangle inequality)} \\
&\leq \eta_t \mathbb{E}[\|w_{t-\tau_{\text{mix}}} - w_r^*\|(2\zeta + \|w_{t-\tau_{\text{mix}}} - w_r^*\|)] && \text{(by the definition of } \zeta) \\
&\leq 3\eta_t B(\tau_{\text{mix}}) && \text{(assuming } d_k \leq B(\tau_{\text{mix}}) \text{ for all } k \in [t], \|w_{t-\tau_{\text{mix}}} - w_r^*\| \leq B(\tau_{\text{mix}}), \text{ and } \zeta \leq B(\tau_{\text{mix}}))
\end{aligned}
$$

$$
\implies \mathbb{E}[T_2] \leq 3 \, \eta_t \, \ln^2(T) \, B(\tau_{\text{mix}})
$$

Combining $T_1, T_2, T_3, T_4$ yields:

$$
e_t \leq \underbrace{(C_1 + 3 + 2C_2)}_{:=C} \, \eta_t \, \ln^2(T) \, B(\tau_{\text{mix}})
$$

$\square$

In addition to $e_t$, we also need to upper bound $\mathbb{E}_{s_t \sim P_\pi^t \mu_0} \left[ \| g_t^r(w_t) - g^r(w_t) \|^2 \right]$.

**Lemma 4.8.** *Assuming* $\| w_k - w_r^* \|^2 \leq B(\tau_{mix}), \forall k \in [t]$, *then we have*

$$\mathbb{E}_{s_t \sim P_\pi^t \mu_0} \left[ \| g_t^r(w_t) - g^r(w_t) \|^2 \right] \leq C' B(\tau_{mix}),$$

*where* $C' = 10(3 + \lambda)^2$.

*Proof.*

$$\mathbb{E}_{s_t \sim P_\pi^t \mu_0} \left[ \| g_t^r(w_t) - g^r(w_t) \|^2 \right]$$
$$\leq \mathbb{E} \left[ 2 \| g_t^r(w_t) \|^2 + 2 \| g^r(w_t) \|^2 \right]. \qquad (\| x - y \|^2 \leq 2 \| x \|^2 + 2 \| y \|^2)$$
$$\leq 2\mathbb{E} \left[ ((2 + \lambda) \| w_t - w_r^* \| + (3 + \lambda)\varsigma)^2 + ((2 + \lambda) \| w_t - w_r^* \|)^2 \right] \qquad \text{(by Lemma 4.2 and Lemma 4.3)}$$
$$\leq \underbrace{10(3 + \lambda)^2}_{:= C'} B(\tau_{\text{mix}}). \qquad \text{(assuming } d_k \leq B(\tau_{\text{mix}}) \text{ for all } k \in [t])$$

$\square$

For the subsequent steps, the proofs for standard TD(0) and regularized TD(0) differ. We first provide the convergence rate for standard TD(0), where the step-size depends on $\omega$, and then show that regularized TD(0) removes this requirement.

### E.1. Standard TD(0)

**Lemma 4.9.** *For the standard TD(0) update, when* $T \geq \max\{3, \frac{1}{\eta_0}, \frac{\ln(4Tm/\eta_0)}{\ln(1/\rho)}\}$, *and* $T$ *is large enough that* $\frac{\ln^2(T)}{T} \leq \frac{1}{2a}$, $\frac{\ln(T)}{T} \leq \frac{1}{b}$, *and* $\ln(T) \geq \max\{a, b\}$, *for a fixed* $t$, *if* $\| w_k - w_r^* \|^2 \leq B(\tau_{mix})$ *for all* $k \leq t$ *and*

$$\eta_0 \leq \frac{(1 - \gamma)\omega}{2 \left[ C \ln^2(T) + C' \right]},$$

*then* $\| w_{t+1} - w^* \|^2 \leq B(\tau_{mix})$, *and hence,* $\| w_k - w_r^* \|^2 \leq B(\tau_{mix})$ *for all* $k \leq t + 1$.

*Proof.* We use the above lemmas and prove the result by induction. We assume that for any $t \geq \tau_{\text{mix}}$, $d_k \leq B(\tau_{\text{mix}})$ for all $k \in [t]$. Now we show that, with an appropriate choice of $\eta_0$, we have $d_{k+1} \leq B$. We continue the expansion in Equation (5) and take expectation with respect to the randomness at iteration $t$:

$$\mathbb{E} \left[ \| w_{t+1} - w^* \|^2 \right]$$
$$\leq \| w_t - w^* \|^2 + 2\eta_t \mathbb{E} \left[ \langle g_t(w_t) - g(w_t), w_t - w^* \rangle \right] + 2\eta_t^2 \mathbb{E} \left[ \| g_t(w_t) - g(w_t) \|^2 \right] + 2\eta_t^2 \| g(w_t) \|^2 + 2\eta_t \langle g(w_t), w_t - w^* \rangle$$
$$\leq \| w_t - w^* \|^2 + 2\eta_t \langle g(w_t), w_t - w^* \rangle + 2\eta_t^2 \| g(w_t) \|^2 + 2C \eta_t^2 \ln^2(T) B(\tau_{\text{mix}}) + 2C' \eta_t^2 B(\tau_{\text{mix}})$$
$$\qquad \text{(using Lemmas 4.7 and 4.8)}$$
$$= \| w_t - w^* \|^2 + 2\eta_t \langle g(w_t), w_t - w^* \rangle + 2\eta_t^2 \| g(w_t) \|^2 + 2 \left[ C \ln^2(T) + C' \right] \eta_t^2 B(\tau_{\text{mix}})$$

We note that $\| w_t - w^* \|^2 + 2\eta_t \langle g(w_t), w_t - w^* \rangle + 2\eta_t^2 \| g(w_t) \|^2$ is similar to the analysis for the mean-path update in Appendix B. We continue the analysis as follows:

$$\mathbb{E} \left[ \| w_{t+1} - w^* \|^2 \right] \leq \| w_t - w^* \|^2 + \left( 16\eta_t^2 - 2(1 - \gamma)\eta_t \right) \| V_{w_t} - V_{w^*} \|^2 + 2 \left[ C \ln^2(T) + C' \right] \eta_t^2 B(\tau_{\text{mix}})$$
$$\qquad \text{(by Lemma 3.1 and Lemma 3.3)}$$

Setting $\eta_0 \leq \frac{1 - \gamma}{16 \ln(T)} < 1$, we can guarantee $\eta_t \leq \frac{1 - \gamma}{16}$, thus

$$\mathbb{E} \left[ \| w_{t+1} - w^* \|^2 \right] \leq \| w_t - w^* \|^2 - (1 - \gamma)\eta_t \| V_{w_t} - V_{w^*} \|^2 + 2 \left[ C \ln^2(T) + C' \right] \eta_t^2 B(\tau_{\text{mix}})$$
$$\leq \| w_t - w^* \|^2 - (1 - \gamma)\eta_t \omega \| w_t - w^* \|^2 + 2 \left[ C \ln^2(T) + C' \right] \eta_t^2 B(\tau_{\text{mix}}) \qquad \text{(by Lemma 3.2)}$$

Under the assumption $d_k \leq B(\tau_{\text{mix}})$ for all $k \in [t]$, and since $(1-\gamma)\eta_t\omega \leq 1$, we have

$$\mathbb{E}\left[\|w_{t+1} - w^*\|^2\right] \leq \left(1 - (1-\gamma)\eta_t\omega + 2\left[C\ln^2(T) + C'\right]\eta_t^2\right)B(\tau_{\text{mix}}).$$

When $\eta_0 \leq \min\left\{\frac{(1-\gamma)\omega}{2\left[C\ln^2(T)+C'\right]}, \frac{1-\gamma}{16\ln(T)}\right\}$, we have $\left(1 - (1-\gamma)\eta_t\omega + 2\left[C\ln^2(T)+C'\right]\eta_t^2\right) \leq 1$.

Furthermore, since $0 < (1-\gamma) \leq 1, 0 < \omega < 1, T \geq 3$, we know that $\eta_0 \leq 1$, and consequently, $(1-\gamma)\eta_t\omega \leq (1-\gamma)\eta_0\omega < 1$, implying that $1 - (1-\gamma)\eta_t\omega > 0$.

Hence, $\left(1 - (1-\gamma)\eta_t\omega + 2\left[C\ln^2(T)+C'\right]\eta_t^2\right) \in (0,1)$, and consequently,

$$\mathbb{E}\left[\|w_{t+1} - w^*\|^2\right] \leq B(\tau_{\text{mix}}),$$

which completes the induction.

Plugging in the value of $C$ and $C'$ with $\lambda = 0$, we have $\frac{(1-\gamma)\omega}{2[C\ln^2(T)+C']} = \frac{(1-\gamma)\omega}{446\ln^2(T)+180}$. Since $\omega \leq 1, \ln^2(T) \geq \ln(T)$, we have $\frac{(1-\gamma)\omega}{446\ln^2(T)+180} \leq \frac{1-\gamma}{16\ln(T)}$. Thus it suffices to have $\eta_0 \leq \frac{(1-\gamma)\omega}{2[C\ln^2(T)+C']}$. $\qquad\square$

The next theorem quantifies the convergence rate of standard TD(0) with exponential step-sizes under Markovian sampling.

**Theorem 4.10.** *The standard TD(0) update with exponential step-sizes $\eta_t = \eta_0\alpha_t$, where $\eta_0 = \frac{(1-\gamma)\omega}{2\left[C\ln^2(T)+C'\right]}$, $\alpha_t = \alpha^t = \frac{1}{T}^{t/T}$, and $T \geq \max\{\frac{1}{\eta_0}, \frac{\ln(4Tm/\eta_0)}{\ln(1/\rho)}\}$, and $T$ is large enough that $\frac{\ln^2(T)}{T} \leq \frac{1}{2a}$, $\frac{\ln(T)}{T} \leq \frac{1}{b}$, and $\ln(T) \geq \max\{a, b\}$, achieves the following convergence rate:*

$$\mathbb{E}\left[\|w_{T+1} - w^*\|^2\right]$$
$$= O\left(\exp\left(-\frac{\omega^2 T}{\ln^3(T)}\right) + \frac{\ln^4(T)}{\omega^2 T}\exp\left(\frac{m}{\ln(1/\rho)}\right)\right),$$

*where $m$ and $\rho$ are related to mixing time as $\tau_{mix} = \frac{\ln(4Tm/\eta_0)}{\ln(1/\rho)}$.*

*Proof.* Continuing the one-step expansion with step-size $\eta_0 \leq \frac{1-\gamma}{16}$ as in Lemma 4.9, we have that, for the absolute constants $C$ and $C'$ defined in Lemma 4.7 and Lemma 4.8 respectively,

$$\mathbb{E}\left[\|w_{t+1} - w^*\|^2\right] \leq \|w_t - w^*\|^2 - (1-\gamma)\eta_t\omega\|w_t - w^*\|^2 + \underbrace{2\left[C\ln^2(T)+C'\right]}_{:=C(T)}\eta_t^2 B(\tau_{\text{mix}})$$

Taking expectation over $t \in [T]$, we have:

$$\mathbb{E}\left[\|w_T - w^*\|^2\right] \leq \|w_0 - w^*\|^2\exp\left(-\eta_0\omega(1-\gamma)\sum_{t=1}^{T}\alpha^t\right)$$
$$+ C(T)B(\tau_{\text{mix}})\eta_0^2\sum_{t=1}^{T}\alpha^{2t}\exp\left(-\eta_0\omega(1-\gamma)\sum_{i=t+1}^{T}\alpha^i\right).$$

This result has the same form as in Section 3. Applying Lemma F.1 and Lemma F.2, we obtain the convergence rate:

$$\mathbb{E}\left[\|w_{T+1} - w^*\|^2\right] \leq \|w_0 - w^*\|^2 \, e \exp\left(-\eta_0\omega(1-\gamma)\frac{\alpha T}{\ln(T)}\right) + \frac{8C(T)B(\tau_{\text{mix}})}{e(\omega(1-\gamma))^2}\frac{\ln^2(T)}{\alpha^2 T}$$
$$= O\left(\exp\left(-\frac{\omega^2 T}{\ln^3(T)}\right) + \frac{\ln^4(T)}{\omega^2 T}\exp\left(\frac{m}{\ln(1/\rho)}\right)\right), \quad \text{(plugging in the values of } \eta_0, B(\tau_{\text{mix}}), C(T))$$

where $m$ and $\rho$ are related to mixing time as $\tau_{\text{mix}} = \frac{\ln(4Tm/\eta_0)}{\ln(1/\rho)}$.

Additionally, for the condition $T \geq \max\{\frac{1}{\eta_0}, 3\}$, when $\eta_0 \leq \frac{(1-\gamma)\omega}{2[C\ln^2(T)+C']}$, $T \geq 1/\eta_0$ implies $T \geq 3$. Thus, it suffices that $T \geq \frac{1}{\eta_0}$. $\qquad\square$

### E.2. Regularized TD(0)

Now we provide the proof for regularized TD(0), and demonstrate that it does not require $\omega$.

**Lemma 4.11.** *For the regularized TD(0) update, when $T \geq \max\{3, \frac{1}{\eta_0}, \frac{\ln(2(2+\lambda)Tm/\eta_0)}{\ln(1/\rho)}\}$, and $T$ is large enough that $\frac{\ln^2(T)}{T} \leq \frac{1}{2a}$, $\frac{\ln(T)}{T} \leq \frac{1}{b}$, and $\ln(T) \geq \max\{a, b\}$, for a fixed t, if $\|w_k - w_r^*\|^2 \leq B(\tau_{mix})$ for all $k \leq t$ and the step-size satisfies*

$$\eta_0 \leq \frac{\lambda}{[C \ln^2(T) + C'] + (8 + 2\lambda^2)},$$

*then $\|w_{t+1} - w_r^*\|^2 \leq B(\tau_{mix})$, and hence, $\|w_k - w_r^*\|^2 \leq B(\tau_{mix})$ for all $k \leq t+1$.*

*Proof.* We use the above lemmas and prove the result by induction. We assume that for any $t \geq \tau_{\text{mix}}$, $d_k \leq B(\tau_{\text{mix}})$ for all $k \in [t]$. Now we show that, with an appropriate choice of $\eta_0$, we have $d_{k+1} \leq B$. We continue the expansion in Equation (5) and take expectation with respect to the randomness at iteration $t$:

$$\mathbb{E}\left[\|w_{t+1} - w^*\|^2\right]$$

$$\leq \|w_t - w^*\|^2 + 2\eta_t \mathbb{E}\left[\langle g_t(w_t) - g(w_t), w_t - w^*\rangle\right] + 2\eta_t^2 \mathbb{E}\left[\|g_t(w_t) - g(w_t)\|^2\right] + 2\eta_t^2 \|g(w_t)\|^2 + 2\eta_t \langle g(w_t), w_t - w^*\rangle$$

$$\leq \|w_t - w^*\|^2 + 2\eta_t \langle g(w_t), w_t - w^*\rangle + 2\eta_t^2 \|g(w_t)\|^2 + 2C \eta_t^2 \ln^2(T) B(\tau_{\text{mix}}) + 2C' \eta_t^2 B(\tau_{\text{mix}})$$
$$\text{(using Lemmas 4.7 and 4.8)}$$

$$= \|w_t - w^*\|^2 + 2\eta_t \langle g(w_t), w_t - w^*\rangle + 2\eta_t^2 \|g(w_t)\|^2 + 2\left[C \ln^2(T) + C'\right] \eta_t^2 B(\tau_{\text{mix}})$$

We note that $\|w_t - w^*\|^2 + 2\eta_t \langle g(w_t), w_t - w^*\rangle + 2\eta_t^2 \|g(w_t)\|^2$ is similar to the analysis for a mean-path update. We continue the analysis as follows:

$$\mathbb{E}\left[\|w_{t+1} - w_r^*\|^2\right] \leq \|w_t - w_r^*\|^2 + \left[2(8 + 2\lambda^2)\eta_t^2 - 2\lambda\eta_t\right] \|w_t - w_r^*\|^2 - 2\eta_t(1 - \gamma)\omega \|w_t - w_r^*\|^2$$
$$+ 2\left[C \ln^2(T) + C'\right] \eta_t^2 B(\tau_{\text{mix}}) \qquad \text{(by Lemma E.1 and Lemma E.2)}$$
$$\leq \left(1 + \left[2(8 + 2\lambda^2)\eta_t^2 - 2\lambda\eta_t\right]\right) \|w_t - w_r^*\|^2 + 2\left[C \ln^2(T) + C'\right] \eta_t^2 B(\tau_{\text{mix}})$$

If $\eta_0 < \frac{1}{2\lambda}$, $\eta_t < \frac{1}{2\lambda}$ and consequently, $2(8 + 2\lambda^2)\eta_t^2 - 2\lambda\eta_t > -1$. Hence, for $\eta_0 \leq \frac{1}{2\lambda}$, $1 + 2(8 + 2\lambda^2)\eta_t^2 - 2\lambda\eta_t > 0$.

Under the assumption $d_k \leq B(\tau_{\text{mix}})$ for all $k \in [t]$, and consequently,

$$\mathbb{E}\left[\|w_{t+1} - w_r^*\|^2\right] \leq \left(1 + 2(8 + 2\lambda^2)\eta_t^2 - 2\lambda\eta_t + 2\left[C \ln^2(T) + C'\right] \eta_t^2\right) B(\tau_{\text{mix}})$$

For $\eta_0 \leq \frac{\lambda}{[C \ln^2(T) + C'] + (8 + 2\lambda^2)}$, $\left(1 + 2(8 + 2\lambda^2)\eta_t^2 - 2\lambda\eta_t + 2\left[C \ln^2(T) + C'\right] \eta_t^2\right) \leq 1$. Hence,

$$\mathbb{E}\left[\|w_{t+1} - w_r^*\|^2\right] \leq B(\tau_{\text{mix}})$$

This completes the induction.

$\square$

We now state the final convergence rate for regularized TD(0) under Markovian sampling.

**Theorem 4.12.** *The regularized TD(0) update with exponential step-size $\eta_t = \eta_0\alpha_t$, where $\eta_0 = \frac{\lambda}{[C \ln^2(T) + C'] + (8 + 2\lambda^2)}$, $\alpha_t = \alpha^t = \frac{1}{T}^{t/T}$, and $T \geq \max\{\frac{1}{\eta_0}, \frac{\ln(2(2+\lambda)Tm/\eta_0)}{\ln(1/\rho)}\}$, and $T$ is large enough that $\frac{\ln^2(T)}{T} \leq \frac{1}{2a}$, $\frac{\ln(T)}{T} \leq \frac{1}{b}$, $\ln(T) \geq \max\{a, b\}$, and $\lambda = 1/\sqrt{T}$, achieves the following convergence rate:*

$$\mathbb{E}\left[\|w_{T+1} - w^*\|^2\right]$$
$$= O\left(\exp\left(-\frac{\omega\sqrt{T}}{\ln^3(T)}\right) + \frac{\ln^4(T)}{\omega^2 T} \exp\left(\frac{m}{\ln(1/\rho)}\right)\right),$$

*where $m$ and $\rho$ are related to mixing time as $\tau_{mix} = \frac{\ln(2(2+\lambda)Tm/\eta_0)}{\ln(1/\rho)}$.*

*Proof.* As in the proof of Lemma 4.11, we obtain that if $\eta_0 \leq \min\left\{\frac{1}{2\lambda}, \frac{1-\gamma}{16\ln(T)}, \frac{\lambda}{[C\ln^2(T)+C']+(8+2\lambda^2)}\right\}$,

$$
\|w_{t+1} - w_r^*\|^2 \leq \|w_t - w_r^*\|^2 + \left[2(8+2\lambda^2)\eta_t^2 - 2\lambda\eta_t\right]\|w_t - w_r^*\|^2 - 2\eta_t(1-\gamma)\omega\|w_t - w_r^*\|^2
$$
$$
+ \underbrace{2\left[C\ln^2(T)+C'\right]}_{:=C(T)}\eta_t^2\,B(\tau_{\mathrm{mix}})
$$

Moreover for $C'' = 10$, since $\eta_t \leq \eta_0$, if $\eta_0 \leq \frac{\lambda}{C''} \leq \frac{2\lambda}{2(8+2\lambda^2)}$, $2(8+2\lambda^2)\eta_t^2 - 2\lambda\eta_t < 0$.

Hence, for $\eta_0 = \min\left\{\frac{1}{2\lambda}, \frac{1-\gamma}{16\ln(T)}, \frac{\lambda}{[C\ln^2(T)+C']+C''}, \frac{\lambda}{C''}\right\}$,

$$
\|w_{t+1} - w_r^*\|^2 \leq (1 - 2\eta_t(1-\gamma)\omega)\|w_t - w_r^*\|^2 + C(T)\eta_t^2 B(\tau_{\mathrm{mix}})
$$

Taking expectations over $t \in [T]$ and recursing,

$$
\mathbb{E}\left[\|w_{T+1} - w_r^*\|^2\right] \leq \|w_1 - w_r^*\|^2 \prod_{t=1}^T (1 - 2\eta_0\alpha^t(1-\gamma)\omega) + C(T)B(\tau_{\mathrm{mix}})\eta_0^2 \sum_{t=1}^T \alpha^{2t} \prod_{i=t+1}^T (1 - 2\eta_0\alpha^t(1-\gamma)\omega)
$$
$$
\leq \|w_1 - w_r^*\|^2 \exp\left(-2\eta_0\omega(1-\gamma)\sum_{t=1}^T \alpha^t\right) + C(T)B(\tau_{\mathrm{mix}})\eta_0^2 \sum_{t=1}^T \alpha^{2t} \exp\left(-2\eta_0\omega(1-\gamma)\sum_{i=t+1}^T \alpha^i\right)
$$

Similar to the proof in Appendix D, applying Lemma F.1 and Lemma F.2 yields:

$$
\mathbb{E}\left[\|w_{T+1} - w_r^*\|^2\right] \leq \|w_1 - w_r^*\|^2\,e\exp\left(-2\eta_0\omega(1-\gamma)\frac{\alpha T}{\ln(T)}\right) + C(T)B(\tau_{\mathrm{mix}})\frac{4}{e^2(\omega(1-\gamma))^2}\frac{\ln^2(T)}{\alpha^2 T}.
$$

Expressing the result in terms of the distance to $w^*$:

$$
\mathbb{E}\left[\|w_{T+1} - w^*\|^2\right]
$$
$$
\leq 2\mathbb{E}\left[\|w_{T+1} - w_r^*\|^2\right] + 2\|w_r^* - w^*\|^2 \qquad\qquad \text{(since } (x+y)^2 \leq 2x^2 + 2y^2)
$$
$$
\leq \|w_1 - w_r^*\|^2\,2e\exp\left(-2\eta_0\omega(1-\gamma)\frac{\alpha T}{\ln(T)}\right) + \frac{8\,C(T)\,B(\tau_{\mathrm{mix}})}{e^2(\omega(1-\gamma))^2}\frac{\ln^2(T)}{\alpha^2 T} + \frac{2\lambda^2\|w^*\|^2}{(1-\gamma)^2\omega^2}. \qquad \text{(by Lemma E.3)}
$$

Setting $\lambda = \frac{1}{\sqrt{T}} < 1$ gives:

$$
\mathbb{E}\left[\|w_{T+1} - w^*\|^2\right] \leq \|w_1 - w_r^*\|^2\,2e\exp\left(-2\eta_0\omega(1-\gamma)\frac{\alpha T}{\ln(T)}\right) + \frac{8\,C(T)\,B(\tau_{\mathrm{mix}})}{e^2(\omega(1-\gamma))^2}\frac{\ln^2(T)}{\alpha^2 T} + \frac{2\|w^*\|^2}{(\omega(1-\gamma))^2 T}
$$
$$
= O\left(\exp\left(-\frac{\omega\sqrt{T}}{\ln^3(T)}\right) + \frac{\ln^4(T)}{\omega^2 T}\exp\left(\frac{m}{\ln(1/\rho)}\right)\right),
$$
$$
\text{(plugging in the values of } \eta_0, B(\tau_{\mathrm{mix}}), C(T))
$$

where $m$ and $\rho$ are related to mixing time as $\tau_{\mathrm{mix}} = \frac{\ln(4Tm/\eta_0)}{\ln(1/\rho)}$.

Additionally, for the condition on $T$, when $\eta_0 \leq \frac{\lambda}{[C\ln^2(T)+C']+(8+2\lambda^2)}$, $T \geq 1/\eta_0$ implies $T \geq 3$. Thus, it suffices that $T \geq \max\{\frac{1}{\eta_0}, \frac{\ln(4Tm/\eta_0)}{\ln 1/\rho}\}$, and that $T$ is large enough that $\frac{\ln^2(T)}{T} \leq \frac{1}{2a}$, $\frac{\ln(T)}{T} \leq \frac{1}{b}$, and $\ln(T) \geq \max\{a, b\}$. $\qquad\square$

## F. Helper Lemmas

**Lemma F.1.**

$$X := \sum_{t=1}^{T} \alpha^t \geq \frac{\alpha T}{\ln(T)} - \frac{1}{\ln(T)}.$$

*Proof.*

$$\sum_{t=1}^{T} \alpha^t = \frac{\alpha - \alpha^{T+1}}{1 - \alpha} = \frac{\alpha}{1 - \alpha} - \frac{\alpha^{T+1}}{1 - \alpha}.$$

We have

$$\frac{\alpha^{T+1}}{1 - \alpha} = \frac{\alpha}{T(1 - \alpha)} = \frac{1}{T} \frac{1}{1/\alpha - 1} \leq \frac{1}{T} \frac{1}{\ln(1/\alpha)} = \frac{1}{\ln(T)},$$

where in the inequality we used Lemma 4 and the fact that $1/\alpha > 1$. Plugging back into $X$ we get

$$X \geq \frac{\alpha}{1 - \alpha} - \frac{1}{\ln(T)} \geq \frac{\alpha}{\ln(1/\alpha)} - \frac{1}{\ln(T)} = \frac{\alpha T}{\ln(T)} - \frac{1}{\ln(T)}.$$

$\square$

**Lemma F.2.** *For $\alpha = \frac{1}{T}^{1/T}$ and any $\kappa > 0$,*

$$\sum_{t=1}^{T} \alpha^{2t} \exp\left(-a \sum_{i=t+1}^{T} \alpha^i\right) \leq \frac{4c \left(\ln(T)\right)^2}{a^2 e^2 \alpha^2 T},$$

where $c = \exp\left(a \frac{1}{\ln(T)}\right)$.

*Proof.* First, observe that,

$$\sum_{i=t+1}^{T} \alpha^i = \frac{\alpha^{t+1} - \alpha^{T+1}}{1 - \alpha}$$

We have

$$\frac{\alpha^{T+1}}{1 - \alpha} = \frac{\alpha}{T(1 - \alpha)} = \frac{1}{T} \cdot \frac{1}{1/\alpha - 1} \leq \frac{1}{T} \cdot \frac{1}{\ln(1/\alpha)} = \frac{1}{\ln(T)}$$

These relations imply that,

$$\sum_{i=t+1}^{T} \alpha^i \geq \frac{\alpha^{t+1}}{1 - \alpha} - \frac{1}{\ln(T)}$$

$$\implies \exp\left(-a \sum_{i=t+1}^{T} \alpha^i\right) \leq \exp\left(-a \frac{\alpha^{t+1}}{1 - \alpha} + a \frac{1}{\ln(T)}\right) = c \exp\left(-a \frac{\alpha^{t+1}}{1 - \alpha}\right),$$

where $c = \exp\left(a\frac{1}{\ln(T)}\right)$. We then have

$$
\sum_{t=1}^{T} \alpha^{2t} \exp\left(-a \sum_{i=t+1}^{T} \alpha^i\right) \leq c \sum_{t=1}^{T} \alpha^{2t} \exp\left(-a \frac{\alpha^{t+1}}{1-\alpha}\right)
$$

$$
\leq c \sum_{t=1}^{T} \alpha^{2t} \left(\frac{2(1-\alpha)}{ea\alpha^{t+1}}\right)^2 \qquad \text{(by Lemma F.3 with } \nu = 2\text{)}
$$

$$
= \frac{4c}{a^2 e^2 \alpha^2} T (1-\alpha)^2
$$

$$
\leq \frac{4c}{a^2 e^2 \alpha^2} T (\ln(1/\alpha))^2
$$

$$
= \frac{4c(\ln(T))^2}{a^2 e^2 \alpha^2 T}
$$

$\square$

**Lemma F.3.** *For all $x, \nu > 0$,*

$$
\exp(-x) \leq \left(\frac{\nu}{ex}\right)^\nu
$$

*Proof.* Let $x > 0$. Define $f(\nu) = \left(\frac{\nu}{ex}\right)^\nu - \exp(-x)$. We have

$$
f(\nu) = \exp\left(\nu \ln(\nu) - \nu \ln(ex)\right) - \exp(-x)
$$

and

$$
f'(\nu) = \left(\nu \cdot \frac{1}{\nu} + \ln(\nu) - \ln(ex)\right) \exp\left(\nu \ln(\nu) - \nu \ln(ex)\right)
$$

Thus

$$
f'(\nu) \geq 0 \iff 1 + \ln(\nu) - \ln(ex) \geq 0 \iff \nu \geq \exp\left(\ln(ex) - 1\right) = x
$$

So $f$ is decreasing on $(0, x]$ and increasing on $[x, \infty)$. Moreover,

$$
f(x) = \left(\frac{x}{ex}\right)^x - \exp(-x) = \left(\frac{1}{e}\right)^x - \exp(-x) = 0
$$

and thus $f(\nu) \geq 0$ for all $\nu > 0$ which proves the lemma. $\square$

# G. Doubling trick

In this section, we introduce a doubling-trick wrapper that removes the need to know the horizon $T$ in advance. We treat our three methods—standard TD(0) with exponential step-size under i.i.d. sampling, standard TD(0) with exponential step-size under Markovian sampling, and regularized TD(0) under Markovian sampling—as black boxes that admit a per-epoch convergence guarantee. Algorithm 1 describes the wrapper, and Theorem G.1 shows that it preserves the rate of the underlying method (stated for regularized TD(0) under Markovian sampling; the other two cases are analogous).

**Notation.** We index the doubling epochs by $s = 0, 1, \ldots, K$. Let $\hat{w}_s$ denote the iterate at the end of epoch $s$, with $\hat{w}_{-1}$ being the user-supplied initial iterate. Epoch $s$ runs the underlying TD update for $T_s := 2^s T_0$ iterations starting from $\hat{w}_{s-1}$, with hyperparameters set as if the horizon were $T_s$; the resulting iterate is $\hat{w}_s$. The total number of TD updates across all $K + 1$ epochs is

$$
T := \sum_{s=0}^{K} T_s = T_0 \left(2^{K+1} - 1\right).
$$

Note that $T_0 \cdot 2^K \leq T \leq 2 \cdot T_0 \cdot 2^K$, so $T$ and the final epoch length $T_K$ agree up to a factor of two.

---

**Algorithm 1** Doubling-trick wrapper for TD.

---

1: **Input:** initial iterate $\hat{w}_{-1}$; base horizon $T_0 \geq 1$; number of doubling epochs $K \geq 0$.
2: **for** $s = 0, 1, \ldots, K$ **do**
3: $\quad T_s \leftarrow 2^s T_0$.
4: $\quad$ Set the TD hyperparameters as if the horizon were $T_s$.
5: $\quad$ Starting from $\hat{w}_{s-1}$, run TD for $T_s$ iterations; let $\hat{w}_s$ be the resulting iterate.
6: **end for**
7: **return** $\hat{w}_K$.

---

**Theorem G.1.** *Consider Algorithm 1 instantiated with regularized TD(0) using exponential step-sizes, where each epoch $s$ uses horizon $T_s = 2^s T_0$, initial step-size $\eta_0$, and decay rate $\alpha_s = (1/T_s)^{1/T_s}$. Suppose $\eta_0$ and $T_0$ satisfy the conditions of Theorem 4.12, so that the per-epoch bound from the proof of Theorem 4.12 applies for every epoch $s$:*

$$\mathbb{E}\left[\|\hat{w}_s - w^*\|^2\right] \leq 2e \|\hat{w}_{s-1} - w_r^*\|^2 \exp\left(-2\eta_0\omega(1-\gamma)\frac{\alpha_s T_s}{\ln(T_s)}\right)$$
$$+ \frac{8\,C(T_s)\,B(\tau_{mix})}{e^2(\omega(1-\gamma))^2}\frac{\ln^2(T_s)}{\alpha_s^2 T_s} + \frac{2\|w^*\|^2}{(\omega(1-\gamma))^2\,T_s}.$$

*Then after the $K+1$ doubling epochs, with total iterations $T = T_0(2^{K+1} - 1)$, the final iterate satisfies*

$$\mathbb{E}\left[\|\hat{w}_K - w^*\|^2\right] = \tilde{O}\left(\exp\left(-\frac{\omega\sqrt{T}}{\ln^3(T)}\right) + \frac{\ln^4(T)}{\omega^2 T}\exp\left(\frac{m}{\ln(1/\rho)}\right)\right).$$

*Proof.* Let $\alpha_0 := (1/T_0)^{1/T_0}$. Since $\alpha_s = (1/T_s)^{1/T_s}$ is increasing in $T_s$ and $T_0 \leq T_s \leq T$, we have $\alpha_0 \leq \alpha_s$ and $\ln(T_s) \leq \ln(T)$. Substituting these into the per-epoch bound from the theorem statement and collecting the resulting $T$-dependent (but $s$-independent) constants,

$$\mathbb{E}\left[\|\hat{w}_s - w^*\|^2\right] \leq 2e \|\hat{w}_{s-1} - w_r^*\|^2 \exp\left(-\tilde{C}_1(T)\,T_s\right) + \frac{\tilde{C}_2(T)}{T_s}, \tag{10}$$

where

$$\tilde{C}_1(T) := \frac{2\eta_0\omega(1-\gamma)\,\alpha_0}{\ln(T)},$$
$$\tilde{C}_2(T) := \frac{8\,C(T)\,B(\tau_{\mathrm{mix}})}{e^2(\omega(1-\gamma))^2}\frac{\ln^2(T)}{\alpha_0^2} + \frac{2\|w^*\|^2}{(\omega(1-\gamma))^2}.$$

Unrolling Equation (10) for $s = 0, 1, \ldots, K$ and using $\sum_{s=0}^K T_s = T$,

$$\mathbb{E}\left[\|\hat{w}_K - w^*\|^2\right] \leq (2e)^{K+1} \|\hat{w}_{-1} - w_r^*\|^2 \exp\left(-\tilde{C}_1(T)\,T\right) + \tilde{C}_2(T)\sum_{s=0}^K \frac{1}{T_s}\exp\left(-\tilde{C}_1(T)\sum_{s'=s+1}^K T_{s'}\right).$$

For the variance sum, separate out the $s = K$ term (whose inner sum over $s'$ is empty):

$$\sum_{s=0}^{K} \frac{1}{T_s} \exp\left(-\tilde{C}_1(T) \sum_{s'=s+1}^{K} T_{s'}\right) = \frac{1}{T_K} + \sum_{s=0}^{K-1} \frac{1}{T_s} \exp\left(-\tilde{C}_1(T) T_0 \left(2^{K+1} - 2^{s+1}\right)\right)$$

$$\leq \frac{1}{T_K} + \exp\left(-\tilde{C}_1(T) T_0 \, 2^K\right) \sum_{s=0}^{K-1} \frac{1}{T_s} \quad \text{(since } 2^{K+1} - 2^{s+1} \geq 2^K \text{ for } s \leq K - 1\text{)}$$

$$\leq \frac{1}{T_K} + \frac{2}{T_0} \exp\left(-\tilde{C}_1(T) T_0 \, 2^K\right) \quad \text{(since } \sum_{s=0}^{K-1} \frac{1}{T_s} = \frac{1}{T_0} \sum_{s=0}^{K-1} \frac{1}{2^s} \leq \frac{2}{T_0}\text{)}$$

$$\leq \frac{2}{T} + \frac{2}{T_0 \, \tilde{C}_1(T) T_0 \, 2^K} \quad \text{(since } T_K \geq T/2 \text{ and } \exp(-x) \leq 1/x\text{)}$$

$$\leq \frac{2}{T} + \frac{4}{T_0 \, \tilde{C}_1(T) \, T}. \quad \text{(since } T_0 \cdot 2^K \geq T/2\text{)}$$

Combining,

$$\mathbb{E}\left[\|\hat{w}_K - w^*\|^2\right] \leq (2e)^{K+1} \|\hat{w}_{-1} - w_r^*\|^2 \exp\left(-\tilde{C}_1(T) T\right) + \frac{2\tilde{C}_2(T)}{T} + \frac{4\,\tilde{C}_2(T)}{T_0 \, \tilde{C}_1(T) \, T}$$

$$= \tilde{O}\left(\|\hat{w}_{-1} - w_r^*\|^2 \exp\left(-\tilde{C}_1(T) T\right) + \frac{\tilde{C}_2(T)}{T}\right),$$

where the $(2e)^{K+1}$ factor is absorbed into the $\tilde{O}$ since $K = \log_2(T/T_0) - O(1)$ is logarithmic in $T$ and $\exp(-\tilde{C}_1(T) T)$ decays super-polynomially. Plugging in the values of $\tilde{C}_1(T)$ and $\tilde{C}_2(T)$ (matching those in the proof of Theorem 4.12 up to constants) yields

$$\mathbb{E}\left[\|\hat{w}_K - w^*\|^2\right] = \tilde{O}\left(\exp\left(-\frac{\omega\sqrt{T}}{\ln^3(T)}\right) + \frac{\ln^4(T)}{\omega^2 T} \exp\left(\frac{m}{\ln(1/\rho)}\right)\right).$$

$\square$

# H. Experimental Setup and Additional Details

This appendix gives the full construction of the synthetic MDP used in Section 5, the procedures for systematically varying $\omega$ and the mixing time, and the per-method hyperparameter settings reported in the main paper.

## H.1. Inverse MDP Construction: From Value Function to Rewards

**Step 1: Feature Matrix.** The feature matrix $\Phi \in \mathbb{R}^{k \times n}$ is constructed with controlled spectral properties (see Appendix H.2), where $k$ is the feature dimension and $n$ is the number of states.

**Step 2: Optimal Parameter Vector.** Generate the optimal parameter vector $w^* \in \mathbb{R}^k$ by sampling each component independently from a standard normal distribution:

$$w^* \sim \mathcal{N}(0, I_k)$$

**Step 3: Value Function.** Define the value function as the linear combination of features weighted by the optimal parameters:

$$v = \Phi^\top w^* \in \mathbb{R}^n$$

By construction, $v$ lies exactly in the column space of $\Phi^\top$, ensuring zero approximation error.

**Step 4: Transition Matrix.** Generate a stochastic transition matrix $P \in \mathbb{R}^{n \times n}$ where each row defines a probability distribution over next states. The construction applies a softmax transformation to random values:

$$P_{ij} = \frac{\exp(Z_{ij})}{\sum_{j'} \exp(Z_{ij'})}$$

where $Z$ is a matrix of i.i.d. standard normal entries.

**Step 5: Expected Reward Vector.** Given the value function $v$ and transition matrix $P$, derive the expected reward vector $r \in \mathbb{R}^n$ by inverting the Bellman equation. The Bellman equation states:

$$v = r + \gamma P v$$

Rearranging:

$$r = v - \gamma P v = (I - \gamma P) v$$

This ensures that $v$ is the true value function under the reward $r$ and transition $P$.

**Step 6: Reward Matrix.** Construct a reward matrix $R \in \mathbb{R}^{n \times n}$ such that $R_{ij}$ represents the reward for transitioning from state $i$ to state $j$, and the expected rewards match $r$. The constraint is:

$$\sum_j P_{ij} R_{ij} = r_i \quad \text{for all } i$$

or equivalently, $\mathrm{diag}(PR^\top) = r$. For each state $i$, this underdetermined linear system is solved via linear programming:

$$\min_{R_{i,:}} \|R_{i,:}\|_1 \quad \text{subject to} \quad P_{i,:} \cdot R_{i,:}^\top = r_i, \quad R_{ij} \in [-2, 2]$$

**Summary.** The construction pipeline is:

$$\Phi \xrightarrow{\text{random}} w^* \xrightarrow{\Phi^\top w^*} v \xrightarrow[\text{independent}]{\text{generate } P} (v, P) \xrightarrow{(I-\gamma P)v} r \xrightarrow{\text{LP}} R$$

This inverse construction guarantees:

1. **Zero approximation error**: The true value function $v = \Phi^\top w^*$ lies exactly in the feature space.

2. **Bellman consistency**: The rewards, transitions, and value function satisfy the Bellman equation exactly.

3. **Controlled problem parameters**: The feature covariance eigenvalues, mixing time, and other quantities can be systematically varied.

### H.2. Adjusting $\omega$

We construct the feature matrix $\Phi \in \mathbb{R}^{k \times n}$ (where $k \le n$) such that the Gram matrix $\Phi \Phi^\top$ has a prescribed minimum eigenvalue $\lambda_{\min}$.

**Step 1: Specify Target Eigenvalues.** Define the target spectrum for the $k \times k$ Gram matrix $G = \Phi \Phi^\top$:

- Set the smallest eigenvalue exactly to $\lambda_{\min}$
- Sample the remaining $k - 1$ eigenvalues uniformly from $[\lambda_{\min}, \lambda_{\max}]$

This yields eigenvalues $\{\sigma_1, \sigma_2, \ldots, \sigma_k\}$ where $\sigma_1 = \lambda_{\min}$.

**Step 2: Construct the Feature Matrix.**

1. Generate a random $k \times k$ orthogonal matrix $U$ via QR decomposition of a Gaussian random matrix.

2. Form the symmetric positive definite square root:

$$L = U \cdot \mathrm{diag}\left(\sqrt{\sigma_1}, \sqrt{\sigma_2}, \ldots, \sqrt{\sigma_k}\right) \cdot U^\top$$

3. Generate an independent random $n \times n$ orthogonal matrix $V$, and extract $Q \in \mathbb{R}^{k \times n}$ as the first $k$ rows of $V$, yielding a matrix with orthonormal rows, i.e., $QQ^\top = I_k$.

4. Define the feature matrix as:

$$\Phi = LQ$$

**Verification.** By construction:

$$\Phi\Phi^\top = (LQ)(LQ)^\top = LQQ^\top L^\top = LL^\top = U \cdot \mathrm{diag}(\sigma_1, \ldots, \sigma_k) \cdot U^\top$$

Thus, the eigenvalues of $\Phi\Phi^\top$ are exactly $\{\sigma_1, \ldots, \sigma_k\}$, with minimum eigenvalue equal to $\lambda_{\min}$.

**Connection to $\omega$.** The quantity $\omega$ is defined as the smallest eigenvalue of the feature covariance matrix:

$$\Sigma = \Phi D \Phi^\top = \sum_{i=1}^{n} d_i \, \phi_i \phi_i^\top$$

where $D = \mathrm{diag}(d_1, \ldots, d_n)$ contains the stationary distribution and $\phi_i$ denotes the $i$-th column of $\Phi$. By controlling the minimum eigenvalue of $\Phi\Phi^\top$, we indirectly control $\omega$: smaller values of $\lambda_{\min}$ yield feature matrices with reduced spectral spread, leading to smaller values of $\omega$.

## H.3. Adjusting the Mixing Rate of the Markov Chain

The second largest eigenvalue magnitude (SLEM) of the transition matrix controls how quickly the Markov chain converges to its stationary distribution.

**Procedure.** Given an initial transition matrix $P$ with stationary distribution $\pi$:

1. Construct the rank-one stochastic matrix $J = \mathbf{1}\pi^\top$, where every row equals $\pi^\top$.

2. Replace $P$ with the convex combination:
$$P_{\mathrm{new}} = (1 - \alpha)P + \alpha J$$
   where $\alpha \in [0, 1)$ is the mixing factor.

**Spectral Analysis.** Both $P$ and $J$ share the leading left eigenvector $\pi$ with eigenvalue 1. The matrix $J$ has all other eigenvalues equal to 0. Therefore, the convex combination satisfies:

- Leading eigenvalue: 1 (unchanged)

- Second largest eigenvalue magnitude: $(1 - \alpha) \cdot \mathrm{SLEM}(P)$

- Stationary distribution: $\pi$ (preserved)

This provides a continuous parameter to control the mixing time while preserving the stationary distribution:

$$\mathrm{SLEM}(P_{\mathrm{new}}) = (1 - \alpha) \cdot \mathrm{SLEM}(P)$$

## H.4. Per-method Settings for the Experiments in Section 5

**i.i.d. sampling (Figure 1).** We compare five step-size schedules. $1/(\omega(t+1)(1-\gamma))$ *(Bhandari):* $\eta_t = \frac{2/((1-\gamma)\omega)}{16/((1-\gamma)^2\omega)+t}$, last-iterate. $1/(\omega(1-\gamma))$ *(Bhandari):* constant $\eta = \omega(1-\gamma)/8$, last-iterate. $1/\sqrt{T}$ *(Bhandari):* constant $\eta = 1/\sqrt{T}$, Polyak averaging of all iterates. *Tail Average (Patil):* constant $\eta = (1-\gamma)/(1+\gamma)^2$, $\ell_2$-regularization $\lambda = 1/\sqrt{N}$, tail averaging over the last $N = \lceil T/2 \rceil$ iterates. *Exp (Ours):* $\eta_t = \frac{1-\gamma}{8} \cdot (1/T)^{t/T}$, last-iterate, no averaging, no projection, parameter-free.

**Markovian sampling (Figure 2).** We compare four methods: constant step-size with projected averaging (Bhandari et al., 2018), constant step-size with Polyak–Ruppert weighted averaging (Mitra, 2025), decreasing step-size $\alpha_k = \alpha/(k+1)^\xi$ with uniform Polyak–Ruppert averaging (Haque et al., 2024), and our regularized exponential step-size. For Bhandari et al. (2018), Mitra (2025), and Ours we sweep the step-size over $\alpha \in \{5e-4, 1e-4, 5e-3, 1e-2, 5e-2, 1e-1, 5e-1, 1\}$ and report the best for each. For Haque et al. (2024) we additionally sweep $\alpha \in \{5e-4, 1e-4, 5e-3, 1e-2, 5e-2, 1e-1, 5e-1, 1, 5, 10, 25, 50, 100\}$ and $\xi \in \{0.5, 0.75, 1\}$, reporting the best combination. We do not compare with Patil et al. (2023), because the data drop technique is not used in practice.

