# OpenReview forum: "Towards Parameter-Free Temporal Difference Learning"
_ICML.cc/2026/Conference — ICML 2026 regular_

### Official Review · Reviewer_Rxzp · 2026-03-14

**Soundness:** 3
**Presentation:** 3
**Significance:** 1
**Originality:** 1
**Overall Recommendation:** 3
**Confidence:** 4

**Summary:**

This paper proposes a version of TD(0) with linear function approximation by using an exponentially decaying step-size schedule, so the algorithm avoids needing hard-to-estimate problem-dependent quantities such as the minimum feature-covariance eigenvalue and, in the Markovian case, the mixing time. It proves finite-time last-iterate convergence guarantees under both i.i.d. and Markovian sampling, and in the Markovian setting it also avoids tricks like projection or iterate averaging.

**Compliance With Llm Reviewing Policy:**

Affirmed.

**Final Justification:**

The authors provide a discussion about increasing the size of T geometrically. That made me add one point to their score. However, overall I think this work is a straightforward extension of the prior work.

**Key Questions For Authors:**

- We know that TD-learning is a special case of contractive stochastic approximation. Given that, what are the main difficulties in analyzing the TD-learning with exponential step size compared to the analysis of Li et. al 2021?
- The authors argue that using exponential step size can ensure a O(1/T) rate of convergence without the knowledge of problem setting and without the need for averaging. I want to understand what is the issue with averaging? We know that averaging can achieve asymptotic optimal rate of convergence along with the optimal asymptotic variance. In that case,  why would we not use averaging? What is the cost in using averaging? Note that another advantage of averaging is asymptotic convergence with out the need to set the time horizon T in advance, something that the setting of your paper does not cover. Also, can you compare the leading term of your convergence result with the leading terms achieved through averaging?
- In the Markovian setting (Section 4), you did not cite the work of Srikant et al, which establishes convergence without the need for averaging or projection. In that work, they construct a Lyapunov function to study the convergence of TD-learning without any changes. Again, including averaging in this setting can achieve the optimal asymptotic convergence rate and variance with out any extra cost and needing the knowledge of the problem setting. Please look at "Tight finite time bounds of two-time-scale linear stochastic approximation with markovian noise" by Haque et al. Note that for Markovian TD-learning with averaging to get the optimal rate of convergence, even regularization that you consider in Section 4.2 is not needed.

**Limitations:**

- The step size requires the knowledge of the final horizon T to be set.
- In the Regularized TD(0) section 4.2, we need the knowledge of \rho to set the T.
- The convergence result in Theorem 4.12 has exponential dependence on the mixing time, which is far too weak.

**Strengths And Weaknesses:**

- Soundness: The results seem to be valid.

- Presentation: The presentation has to be updated, such that the main results are presented first, and then in the later parts of the paper (maybe a separate section named proof sketch) present all the lemmas used for the proof. Please also remove the colored text, and just specify any term with an underbrace.

Significance: This is not a significant contribution. As mentioned in my comment below, I believe this result is a special case of the result in Li et al.

Originality: I do not believe the choice of exponential step size is neither theoretically, nor experimentally reasonable.

---

> ### Author Rebuttal · Authors · 2026-03-31
>
> We address the reviewer's main concerns below. We have done some basic experiments and added the results to
>
> > Comparison with Li et al.\ (2021)
>
> **Li et al. (2021) analyze exponential step-sizes for smooth, strongly convex objectives under i.i.d. noise. TD(0) does not fit this framework**: it uses a semi-gradient (not the gradient of any fixed objective), and the key contraction is in value-space $\|\cdot\|_D$-norm via a one-point strong-monotonicity lemma (Lemma~3.1), not standard strong convexity.
>
> Furthermore, Li et al. cover only the i.i.d. setting; our Markovian analysis requires controlling an additional dependence-bias term $g_t(w_t) - g(w_t)$ via an induction argument with no analogue in their work. Our contribution is therefore not a corollary of Li et al.; we will add a clearer technical comparison in the final version of the paper.
>
> > Issue with averaging
>
> Our claim is not that averaging should never be used, but that the last iterate is what practitioners use by default, and hence our analysis is more aligned with practice. For example, in large-scale RL applications such as LLM alignment, a single model can occupy 30--400 GB of memory, making it impractical to store a running average of the value model alongside the policy. Last-iterate convergence avoids this overhead entirely.
>
> More importantly, as shown in our Table~1 and established in Mitra (2025), optimal averaging-based rates under Markovian sampling require knowledge of $\tau_{\mathrm{mix}}$ and $\omega$ to set the step-size or the number of burn-in steps. Our method achieves last-iterate convergence under Markovian sampling without knowing $\tau_{\mathrm{mix}}$ (and without $\omega$ in the regularized case), which is precisely the parameter-free property we claim.
>
> Regarding the leading term: comparing our rate directly with averaging-based methods is not entirely fair, since our method is problem-independent while theirs are not. For example, Mitra (2025) achieves $\widetilde{O}(\exp(-\omega^2 T / \tau_{\mathrm{mix}}) + \sigma^2 \tau_{\mathrm{mix}} / (\omega^2 T))$ with Polyak--Ruppert averaging, which has better mixing-time dependence (linear vs.\ our exponential), but requires knowledge of both $\omega$ and $\tau_{\mathrm{mix}}$ to set the step-size.
> Our regularized TD achieves a comparable $\widetilde{O}(1/T)$ rate without knowing either quantity, and the additional logarithmic factors are a standard cost of last-iterate analysis (also appearing in Vaswani et al. 2022 for strongly-convex minimization).
>
> > Dependence on $T$
>
> The dependence on $T$ can be removed via a standard doubling trick: run the algorithm in epochs $k \in [K]$ of geometrically increasing length $T_k = 2^k$, warm-starting each epoch from the last iterate of the previous one. We use the main theorem (e.g. Thm 3.4 in the iid case) for each epoch (with its own exponential schedule). Note that the total number of updates satisfies $\sum_{k=1}^{K} T_k = O(T)$. In the Markovian case, each epoch should be long enough and satisfy the conditions in Thm 4.12. This adds a constant overhead that does not change the final rate. In particular, we can prove the following theorem for the resulting algorithm.
>
> **Theorem:** Consider the doubling trick instantiated with regularized TD(0) using exponential step-sizes, where each epoch $s$ uses horizon $T_s = 2^s T_0$, initial step-size $\eta_0$, and decay rate $\alpha_s = (1/T_s)^{1/T_s}$.
> Suppose that $\eta_0$ and $T_0$ satisfy the conditions of Theorem 4.12. Then after $K+1$ epochs with total iterations $T = T_0 \cdot 2^K$, the final iterate satisfies
> $$\mathbb{E}\left[ || w_T - w^* ||^2\right] = \tilde O \left(\exp \left(-\sqrt{T}\right) + \frac{1}{T}\right).$$
>
> We will add this result to the final version of the paper.
>
> > Comparison with Srikant & Ying (2019)
>
> We did not explicitly compare to Srikant & Ying (2019) because they have the same limitations as in Mitra et al (2025). In particular, their step-size requires knowledge of $\tau_{\mathrm{mix}}$. Knowing the mixing time is often impractical, and removing this requirement is a central contribution of our work.
>
> > Comparison to Haque et al
>
> Similar to our result in Thm 4.12, Thm 4.2 in Haque et al. shows the convergence of linear TD without requiring the knowledge of problem-dependent constants for setting the step-size. While Haque et al attain the desired convergence for the average iterate, we use an additional regularization to attain the more practical last-iterate convergence. While both works derive an $O(1/T)$ rate, our result has an additional $O(\ln(T))$ term in the leading term. On the other hand, we have an explicit dependence on the fast-decaying transient term and the mixing time, whereas this dependence is not clear for Haque et al. Finally, we note that the analysis in Haque et al uses tools from stochastic approximation, whereas our proof uses standard convex optimization techniques. We will add this comparison to the revised version.

---

> > ### Author Rebuttal · Reviewer_Rxzp · 2026-04-04
> >
> > Regarding Mitra 2025, they only consider constant step size. An alternative is to use a step size of the form $\alpha_t=\alpha/t^\xi$ for some $\xi \in (0.5,1)$, and then average the iterates. I believe in that case, even in the Markovian noise setting, you would not need knowledge of the $\tau_{mix}$ and $\omega$. You can refer to "Tight finite time bounds of two-time-scale linear stochastic approximation with markovian noise" for the analysis of Markovian linear SA with averaging. Also, note that Haque et al achieve a tight leading term.
> > However, given the extra discussion on the geometrically increasing $T_k$, I will increase my score by one.

---

> > > ### Author Response · Authors · 2026-04-06
> > >
> > > We thank the reviewer for increasing their score and for the constructive follow-up.
> > >
> > > Regarding Mitra 2025, we agree that using the polynomial step-size $\alpha_t = \alpha/t^{\xi}$ with $\xi \in (0.5, 1)$ with iterate averaging and following the proof in Haque et al, 2025 is a viable approach for achieving parameter-free rates under Markovian noise. However, we emphasize that such an approach inherently yields average-iterate convergence, whereas our method provides last-iterate guarantees.
> > >
> > > From a practical perspective (as discussed in the rebuttal), the last iterate is used by default and avoids the memory overhead of storing running averages (which can be prohibitive in large-scale RL). Furthermore, as per the suggestion of Rev. 9MGt, we have conducted some basic experiments (see https://anonymous.4open.science/r/TD_anonymous-348C for details) to evaluate the proposed method and compare it to existing works. **Our results show that in both the iid and Markovian settings, our proposed method is either competitive or better than the baselines including those that employ iterate averaging.**
> > >
> > > Finally, we note that from a theoretical optimization perspective, it is challenging to obtain last-iterate guarantees even when the average-iterate results are well established. For example, for SGD in the standard iid setting, even though obtaining the average iterate convergence is a textbook result, proving the guarantee for the last iterate was an open problem [1] and required different techniques [2].
> > >
> > > We hope this clarification helps the reviewer better contextualize our contributions, and reconsider their assessment of our paper.
> > >
> > > [1] Shamir et al, Is Averaging Needed for Strongly Convex Stochastic Gradient Descent?
> > >
> > > [2] Shamir \& Zhang, Stochastic Gradient Descent for Non-smooth Optimization: Convergence Results and Optimal Averaging Schemes

---

### Official Review · Reviewer_63Wf · 2026-03-15

**Soundness:** 3
**Presentation:** 3
**Significance:** 3
**Originality:** 3
**Overall Recommendation:** 5
**Confidence:** 3

**Summary:**

This paper presents parameter free methods for temporal difference learning under linear function approximation, and obtains near optimal bias-variance tradeoffs on the final iterate by leveraging an exponential learning rate schedule without depending on potentially unknown problem dependent constants.

**Compliance With Llm Reviewing Policy:**

Affirmed.

**Key Questions For Authors:**

1. Could the authors describe the implications of having a rank deficient feature matrix (weakening assumption 2.2) on their results. Generally, I guess the sense it we dont really get per-step constant contraction in these operators but I am curious what is the implication on the use of this class of learning rate schedules.
2. The other question I had was wrt designing anytime algorithms that do not require the knowledge of the end time T -- what are the impediments to design a doubling style algorithm that avoids this dependence on T?

**Limitations:**

yes

**Strengths And Weaknesses:**

Soundness: the paper appears technically sound and presents a reasonably important contribution.

Presentation: Reasonably well written.

Significance: Advances our understanding of a very basic algorithm

Originality: I believe the contributions appear novel but I should also claim that I dont know the relevant recent works in this area.

---

> ### Author Rebuttal · Authors · 2026-03-31
>
> We thank the reviewer for their positive review and helpful feedback. We address their concerns below. We have also done some basic experiments and added the results to
>
> > Rank deficient feature matrix
>
> If $\Phi$ is rank deficient, then $\Sigma = \Phi^\top D \Phi$ can be singular, the fixed point $w*$ is not unique, and Assumption~2.2 fails. However, we believe the results extend naturally to this case. From Eq. (1), the TD update direction $g_t(w_t)$  is always a scalar multiple of $\phi(s_t)$, so $w_t$ only ever moves within the column space of $\Phi^\top$. Directions in the null space of $\Phi$ are never updated and do not affect the value function $V_w(s) = w^\top \phi(s)$. Therefore, the entire analysis can be restricted to the column space of $\Phi^\top$, where $\Sigma$ is positive definite and $\omega$ is naturally replaced by the smallest non-zero eigenvalue of $\Sigma$. This is analogous to optimization with a rank-deficient Hessian, where gradient descent operates only in the column space, and convergence is governed by the smallest non-zero eigenvalue.
>
> Under this restriction, the value function $V(w*)$ remains unique even though $w*$ is not, and the contraction step $|| V_{w_1} - V_{w_2} ||^2_D = || w_1 - w_2 ||^2_\Sigma \geq \omega_{+} \|w_1 - w_2\|^2$ holds with $\omega_{+}$ denoting the smallest non-zero eigenvalue of $\Sigma$. We will formalize this extension in the revision.
>
> > Dependence on $T$ and anytime algorithms
>
> The dependence on $T$ can be removed via a standard doubling trick: run the algorithm in epochs $k \in [K]$ of geometrically increasing length $T_k = 2^k$, warm-starting each epoch from the last iterate of the previous one. We use the main theorem (e.g. Thm 3.4 in the iid case) for each epoch (with its own exponential schedule). Note that the total number of updates satisfies $\sum_{k=1}^{K} T_k = O(T)$. In the Markovian case, each epoch should be long enough and satisfy the conditions in Thm 4.12. This adds a constant overhead that does not change the final rate. In particular, we can prove the following theorem for the resulting algorithm.
>
> **Theorem:** Consider the doubling trick instantiated with regularized TD(0) using exponential step-sizes, where each epoch $s$ uses horizon $T_s = 2^s T_0$, initial step-size $\eta_0$, and decay rate $\alpha_s = (1/T_s)^{1/T_s}$.
> Suppose that $\eta_0$ and $T_0$ satisfy the conditions of Theorem 4.12. Then after $K+1$ epochs with total iterations $T = T_0 \cdot 2^K$, the final iterate satisfies
> $$\mathbb{E}\left[ || w_T - w^* ||^2\right] = \tilde O \left(\exp \left(-\sqrt{T}\right) + \frac{1}{T}\right).$$
>
> We will add this result to the final version of the paper.

---

> > ### Author Rebuttal · Reviewer_63Wf · 2026-04-04
> >
> > I assume for the rank deficient case, the contraction rate does slow down right? And, the assumption here is that one can make increasing amounts of progress handling the bottom sub-spaces as we run the algorithm for longer, for instance, for any specific T, one can make progress on an effective number of dimensions with eigenvalues $O(1/T)$ if i understand right. Getting this sketched out would probably give us the result for the infinite dimensional case (e.g. the kernel version).

---

> > > ### Author Response · Authors · 2026-04-06
> > >
> > > We thank the reviewer for their continued engagement. The reviewer's understanding is correct on both points. Indeed, the contraction rate is slower along directions corresponding to smaller eigenvalues of $\Sigma$, and the overall rate is governed by $\omega_+$, the smallest non-zero eigenvalue. The algorithm makes meaningful progress (i.e. bias becomes negligible relative to the variance floor) primarily along directions with $\lambda_i \geq \Omega(\mathrm{polylog}(T)/T)$. With standard assumptions on the spectral decay of the kernel operator (e.g., polynomial or exponential decay), it should be possible to extend these results to the kernel setting. We thank the reviewer for pointing out this interesting connection, and will include a discussion in the final version of the paper.

---

### Official Review · Reviewer_9MGt · 2026-03-22

**Soundness:** 3
**Presentation:** 3
**Significance:** 2
**Originality:** 3
**Overall Recommendation:** 4
**Confidence:** 3

**Summary:**

This paper studies parameter free TD learning.
Specifically, the main idea is to use exponentially decaying step sizes, and in the Markovian setting, a regularized variant, to remove the need for hard-to-estimate problem-dependent constants in algorithm design.
The paper provides finite-time last-iterate guarantees for a near-standard TD method under both i.i.d. and Markovian sampling, with the i.i.d. result achieving the optimal bias-variance trade-off up to logarithmic factors and the Markovian result offering a comparable extension.

**Compliance With Llm Reviewing Policy:**

Affirmed.

**Final Justification:**

All my questions are resolved. Given the new empirical experiment, I raised my score from 3 to 4.

**Key Questions For Authors:**

1. Could the step-size schedule get rid of the dependence of T?
2. Is there a lower bound for parameter free TD?
3. Do the authors believe that exponential scheduling and regularization are fundamentally necessary for parameter-free last-iterate TD guarantees, or are they mainly artifacts of the current proof?

**Limitations:**

yes

**Strengths And Weaknesses:**

**Strengths**
1. The theoretical contribution for i.i.d. sampling is solid. The authors provide the first last-iterate convergence rate for TD that achieves the optimal bias-variance trade-off, without the need of any prior parameter information.
2. The paper is well-written. The proof is clear and easy to follow.

**Weakness**
1. The proposed step-size schedule still requires prior knowledge of the total horizon T. In particular, the exponential schedule is defined explicitly as a function of T, so the method is not fully horizon-free or anytime, which weakens the practical meaning of being parameter-free.
2. In the Markovian setting, the bound contains an exponential dependence on the mixing-related quantity, which is significantly weaker than the linear dependence achieved in prior work (as the authors acknowledge this limitation themselves). This makes the Markovian guarantee less compelling despite the attractive last-iterate and parameter-free properties.
3. The paper lacks an ablation or empirical study showing that its main design choices—especially the exponentially decaying step size and the regularized variant in the Markovian setting—lead to practical benefits beyond the theory.

---

> ### Author Rebuttal · Authors · 2026-03-31
>
> We thank the reviewer for their review and helpful feedback. We address their concerns below.
>
> > Dependence on $T$
>
> The dependence on $T$ can be removed via a standard doubling trick: run the algorithm in epochs $k \in [K]$ of geometrically increasing length $T_k = 2^k$, warm-starting each epoch from the last iterate of the previous one. We use the main theorem (e.g. Thm 3.4 in the iid case) for each epoch (with its own exponential schedule). Note that the total number of updates satisfies $\sum_{k=1}^{K} T_k = O(T)$. In the Markovian case, each epoch should be long enough and satisfy the conditions in Thm 4.12. This adds a constant overhead that does not change the final rate. In particular, we can prove the following theorem for the resulting algorithm.
>
> **Theorem:** Consider the doubling trick instantiated with regularized TD(0) using exponential step-sizes, where each epoch $s$ uses horizon $T_s = 2^s T_0$, initial step-size $\eta_0$, and decay rate $\alpha_s = (1/T_s)^{1/T_s}$.
> Suppose that $\eta_0$ and $T_0$ satisfy the conditions of Theorem 4.12. Then after $K+1$ epochs with total iterations $T = T_0 \cdot 2^K$, the final iterate satisfies
> $$\mathbb{E}\left[ || w_T - w^* ||^2\right] = \tilde O \left(\exp \left(-\sqrt{T}\right) + \frac{1}{T}\right).$$
>
> We will add this result to the final version of the paper.
>
> > Exponential dependence on mixing time
>
> We acknowledge this limitation in the paper. We believe that this a technical artifact of our current analysis rather than a fundamental barrier, and expect it can be improved in future work. *We emphasize that ours is the first attempt at parameter-free last-iterate TD under Markovian sampling, and our focus is to establish last-iterate, non-averaged, projection-free convergence without tuning with $\tau_{\mathrm{mix}}$ or (in the regularized case) $\omega$. It is important to note that this has not been simultaneously achieved by any prior work.*
>
> > Empirical Study
>
> In order to alleviate the reviewer's concerns, we have conducted some basic experiments in both the iid and Markovian setting, and have added the corresponding results to. Please see the settings and results at https://anonymous.4open.science/r/TD_anonymous-348C.
>
> > Lower bound for parameter-free TD:
>
> With respect to the number of iterations, our upper bound matches the standard $\Theta(1/T)$ dependence for smooth, strongly convex stochastic minimization [1]. With respect to $\omega$, as shown in Table~1, all existing methods that do not assume the knowledge of $\omega$ incur a quadratic $O(\sigma^2/(\omega^2 T))$ dependence on $\omega$ in the variance term (Samsonov et al., 2024; Patil et al., 2023; and our result), whereas methods that assume knowledge of $\omega$ achieve a linear $O(\sigma^2/(\omega T))$ dependence (Bhandari et al., 2018, with step-size $O(1/(\omega(t+1)))$). This consistent gap across all known upper bounds suggests an inherent price of adaptivity to unknown $\omega$. A formal lower bound establishing this separation remains an open problem, and we will state this explicitly in the final version of the paper.
>
> > Are exponential scheduling and regularization necessary?
>
> No, we do not claim this is the only approach towards developing a parameter-free algorithm. Our results show they are sufficient to obtain non-averaged last-iterate guarantees without problem-dependent knowledge. Other techniques can also yield parameter-free rates under different assumptions, for example, AdaGrad-style step sizes achieve an $O(1/\sqrt(T))$ in the convex (non-strongly-convex) setting without knowledge of problem-dependent constants. Whether such ideas extend to TD, or whether it is possible to design an unregularized parameter-free Markovian TD algorithm, are interesting open questions that we will highlight.
>
> [1] Nguyen et al, Tight dimension independent lower bound on the expected convergence rate for diminishing step sizes in sgd

---

> > ### Author Rebuttal · Reviewer_9MGt · 2026-04-04
> >
> > All my questions are solved. Given the new emperical results, I am happy to raise my score.

---

> > > ### Author Response · Authors · 2026-04-06
> > >
> > > We are glad that all your concerns have been addressed. Since the discussion period is nearing its end, we kindly request the reviewer to finalize the updated score at their earliest convenience. Thank you again for the constructive and helpful review.

---

### Decision · Program_Chairs · 2026-04-30

**Decision:**

Accept (regular)

**Comment:**

The reviewers agreed that the paper provides a solid theoretical advancement by establishing the first last-iterate convergence rates for TD learning that achieve an optimal bias-variance trade-off without prior knowledge of problem-dependent constants. During the discussion, the authors successfully addressed concerns regarding the dependence on the time horizon $T$ by introducing a doubling trick and provided empirical results that demonstrate the practical utility of their approach. While some reviewers noted that the Markovian bound's exponential dependence on mixing time is weaker than existing averaging-based results, they acknowledged this as a common challenge in last-iterate analysis and found the overall contribution, particularly the avoidance of memory-intensive averaging, highly relevant to the ICML community and large-scale reinforcement learning practitioners.